

# Correlated observation error models for assimilating all-sky infrared radiances

Alan J. Geer[1]

[1]ECMWF, Shinfield Park, Reading RG2 9AX

**Correspondence:** A.J. GEER (alan.geer@ecmwf.int)

**Abstract.** The benefit of hyperspectral infrared sounders to weather forecasting has been improved with the representation of interchannel correlations in the observation error model. The aim is now to assimilate these observations in all-sky conditions. However, in cloudy skies, observation errors exhibit much stronger interchannel correlations, as well as much larger variances, compared to clear sky conditions. An observation error model is developed to represent these effects, building from the symmetric error models developed for all-sky microwave assimilation. The combination of variational quality control with correlated errors is also introduced. The new error model is tested in all-sky assimilation of 7 water vapour sounding channels from the Infrared Atmospheric Sounding Interferometer (IASI). However, its initial formulation degrades both tropospheric and stratospheric analyses. To explain this the eigendeparture and eigenjacobian are introduced as a way of understanding the effect of correlated observation errors in data assimilation. The trailing eigenvalues can be problematic because they strongly amplify high-order harmonic combinations of the water vapour channels, which could have at least three consequences: first, if there are small inter-channel biases, these can be greatly amplified; second, the trailing eigenjacobians map onto features resembling gravity waves that the data assimilation may not be able to handle; finally, these harmonic combinations can amplify trace sensitivities, for example revealing a strong upper stratospheric sensitivity over high cloud in what are usually mid- to upper-tropospheric water vapour channels. The most likely explanation is the sensitivity to gravity wave features that are present in the observations but hard for the data assimilation to handle. After reducing the sensitivity to the trailing eigenjacobians, the new error covariance matrix gives good results in all-sky infrared assimilation.

*Copyright statement.* The author's copyright for this publication is transferred to ECMWF

## 1 Introduction

Geophysical quantities are inferred from indirect observations (such as satellite radiances) using techniques ultimately derived from Bayes theorem. This requires a representation of the error in the prior state and in the observations. Especially in meteorological data assimilation, accurate modelling of the prior or 'background' error is critical (e.g. Bannister, 2008a, b) and the need to improve this has led to major algorithmic developments like the move to hybrid and ensemble data assimilation (e.g. Bonavita et al., 2012; Houtekamer and Zhang, 2016). But in contrast to the sophistication of modern background error




models, observation errors have usually been represented by a single, globally constant standard deviation. This is increasingly recognised as inadequate since observation errors do not just account for the instrument noise, but for errors in the observation operator (e.g. the radiative transfer model that links the state and the observed radiance) and for representation errors (e.g. Janjić et al., 2017). All these error sources can be correlated in time and space, and between satellite channels, and their error variances and correlations can vary greatly depending on the meteorological situation.

Recently many numerical weather prediction (NWP) centres have started to represent observation error with more sophistication. For the assimilation of hyperspectral infrared (IR) sounder radiances in clear-sky conditions, the representation of inter-channel error correlations has improved the skill of operational weather forecasts (Weston et al., 2014; Bormann et al., 2016; Eresmaa et al., 2017; Campbell et al., 2017). For the assimilation of microwave radiances in all-sky conditions, observation error models have needed situation-dependence, representing errors that are smaller in clear-sky conditions and larger in the presence of cloud and precipitation (Geer and Bauer, 2011; Geer et al., 2018a). Along with other developments, this has allowed all-sky microwave assimilation to provide significant gains in forecast skill (Geer et al., 2017). To develop the assimilation of hyperspectral IR radiances in all-sky conditions, it is likely that both inter-channel error correlations and situation dependence will be required. Indeed even for all-sky microwave observations, inter-channel error correlations are present and become much stronger in the presence of cloud (Bormann et al., 2011) although this has been ignored so far. Hence this study aims to find an observation error model that can include both interchannel error correlations and situation-dependence as a function of cloud amount.

The first problem of observation error modelling is to estimate the observation error covariances. These can be inferred from the covariance of background departures ($d$) which is on average equal to the sum of background and observation errors in observation space, $E(dd^T) = \mathbf{HBH^T} + \mathbf{R}$. Here $E()$ is the expectation operator and it has been assumed there are no correlations between background and observation errors. A range of techniques have been proposed to separate the observation errors $\mathbf{R}$ from the background errors $\mathbf{HBH^T}$. If the observation errors have no spatial correlations, then the Hollingsworth and Lönnberg (1986) spatial separation technique is appropriate. If an estimate of the background error is available, it can be subtracted (e.g Bormann and Bauer, 2010). Alternatively (Desroziers et al., 2005) the covariance of background and analysis departures is equal to the observation error in a data assimilation system where the errors are already correctly specified. This latter technique is widely used (e.g. Stewart et al., 2014; Waller et al., 2016a) and it has been the starting point for the observational error covariance matrices used for hyperspectral IR data assimilation at operational centres.

The estimation of situation-dependent error variances for all-sky microwave assimilation has followed a different approach (Geer and Bauer, 2011; Geer et al., 2018a). A piecewise linear error model is fitted, as a function of a 'symmetric' cloud proxy variable, to the standard deviation of the background departures $d$. Hence the error model is fitted to the sum of observations and background errors (the total error). Originally the error model allowed for a scaling factor, to be estimated by trial and error, that was supposed to remove the error variance due to the background errors. In practice the best scaling factor was 1, i.e. no scaling at all and current practice is to provide observation errors that are equal in size to the total errors (e.g. Zhu et al., 2016; Kazumori et al., 2016). Because of the limited predictability of cloud and precipitation at small scales (e.g. Fabry and Sun, 2010) total errors in cloudy situations are dominated by large displacement and intensity errors in forecasted cloud and




precipitation, often imprecisely known as 'mislocation error'. While this might be thought part of the background error, most weather centres use some variant of four-dimensional assimilation, in which a short model forecast is used to map from the assimilation control variables to the atmospheric state at the observation location. The error in this forecast hence belongs in the observation error, unless otherwise represented as model error. This error is also often seen as an error of representation, even if

it does not come from the mismatch in scales between the observation and the model that is more usually called representation error (see Janjić et al., 2017; Geer et al., 2018a).

Since all-sky microwave assimilation has been most successful where $\tilde{\mathbf{R}} \simeq \mathbf{HBH^T} + \mathbf{R}$ (using a tilde to distinguish an assumed error model from the true errors) this suggests that in cloud and precipitation, $\|\mathbf{HBH^T}\| << \|\mathbf{R}\|$, i.e. that observation errors dominate. An alternative hypothesis would be that observation errors needed to be inflated because of other subopti-

malities, most likely relating to the observation error correlations that are still not modelled in all-sky assimilation. However, Harnisch et al. (2016) used the spread of an ensemble of forecasts to estimate the background errors $\mathbf{HBH^T}$ in cloudy conditions, finding that in many cases they are around a third the size of the total error standard deviation (or a ninth of the error variance). This result could have come from a lack of spread in the ensemble, but if not it supports the dominance of observation error in cloudy and precipitating situations.

The current work will follow common practice in all-sky assimilation by fitting an observation error model directly to the covariances of the background departures, assuming that background error is relatively small. This is justified both by the previous success of this approach and by the lack of suitable alternatives. First the Hollingsworth and Lönnberg (1986) approach is ruled out because the mislocation error is spatially correlated. Second, a good estimate of the background error in cloudy situations is not available at ECMWF. Although an ensemble of data assimilations is available (Bonavita et al., 2012)

from which to compute a spread of all-sky background departures, there are some strange features in its estimates of $\mathbf{HBH^T}$ for any observation with a situation-dependent error model. This unresolved issue has been present for many years and it has prevented the use of the EDA ensemble statistics to support the development of all-sky assimilation at ECMWF.

One other option for estimating observation error is the Desroziers et al. (2005) approach but it has not been used here either. First, idealised theoretical studies have shown limitations to its ability to identify the true observation error (e.g. Waller

et al., 2016b; Ménard, 2016). Second, the diagnosed error covariance matrices have never been used in operational assimilation without additional error inflation. Bormann et al. (2016) inflated error standard deviations by a factor 1.75, determined by trial and error, Weston et al. (2014) inflated all eigenvalues of the error covariance matrix, and Campbell et al. (2017) used only the diagnosed error correlations, sticking with pre-existing error variance estimates that were much larger than those diagnosed. Bormann et al. (2015) argue that this does not necessarily invalidate the observation error estimates if inflation is necessary

to address other remaining suboptimalities, such as the lack of treatment for temporal and spatial correlations. However at minimum these estimates are not immediately applicable in real systems without time-consuming trial-and-error adjustment. Third, when applied to humidity sounding channels the Desroziers et al. (2005) technique produces results that are not yet fully understood. Different to the temperature channels it diagnoses observation error standard deviations that are much smaller than the standard deviation of background departures (e.g. Bormann and Bauer, 2010; Waller et al., 2016a) but much larger than

the instrument noise (Bormann et al., 2016). Weston et al. (2014) explained this as representation error from scale mismatches,



observing that it gets worse with coarser model resolution. Nevertheless, in the current work, the best justification to use the covariance of the background departures as an estimate of the observation error, is that in clear-skies this covariance matrix is nearly identical to the 1.75 times inflated diagnosed error covariance matrix of Bormann et al. (2016), at least for humidity sounding channels. Hence just starting from the covariances of the background departures saves work.

This study has been performed as part of wider developments towards the assimilation of IR radiances in all-sky conditions at ECMWF, to be reported by Geer et al. (2018b). Understanding how best to implement a correlated, situation-dependent error model for all-sky assimilation was the final step in getting this working. Section 2 will provide more details of the all-sky IR developments alongside a general description of the ECMWF forecasting system in which this work has been performed. As a first step the technique has been applied to the mid- and upper-tropospheric water vapour channels of Infrared Atmospheric

Sounding Interferometer (IASI) which have smaller errors and more linear sensitivity to the model state, making them more amenable to data assimilation than temperature-sounding or window channels (Chevallier et al., 2004). Section 3 gives a mathematical overview of the place of the error covariance matrices in data assimilation and the importance of the eigenvector representation of these matrices. In particular, it introduces the concepts of *eigendepartures* and *eigenjacobians* which mirror their parent concepts, but involve projection onto the uncorrelated eigenbasis of the error covariance matrix. These diagnostics

give great insight into what is going on when an interchannel error correlation matrix is used in data assimilation. This section also shows how error correlations and situation dependence can be combined, by using a symmetric all-sky error inflation model to inflate the leading eigenvector of the error covariance matrix. This provides an error covariance matrix that resembles the existing clear-sky error model in clear-sky situations (Bormann et al., 2016) but gives larger error variances and larger interchannel error correlations in cloudy situations. The proposed error covariance model is tested in the ECMWF system in

Sec. 4, initially with poor results, but a key realisation was the role of the trailing eigenvalues in amplifying information that the assimilation system may not handle properly, such as biases, gravity waves, and trace sensitivities to the stratosphere. Hence the most successful error model reduces the weight given to the trailing eigenvectors, following earlier studies (Weston et al., 2014; Bormann et al., 2016), but with a broader understanding of why the trailing eigenvectors can be so problematic.

## 2   Methods

### 2.1   Data assimilation framework

ECMWF operates an NWP system with the aim of predicting weather globally for the medium-range and beyond (day 3 onwards). Initial conditions for the forecast are produced by 4D-Variational data assimilation (4D-Var, Rabier et al., 2000) which combines a 12 h background forecast with the observations available within either a 12 h or 6 h assimilation window. The 'delayed cutoff' 12 h window is used to create the background forecasts for the next assimilation windows; the 'early delivery'

6 h window is used to initialise the main forecasts using the most recently available observations, but does not contribute to the next background. An ensemble of data assimilations and an ensemble of forecasts are also run to provide dynamically-varying background errors and estimates of forecast error.



The forecast model is run at TCo1279 horizontal resolution (around 8–9 km) and with 137 terrain-following vertical levels. Cloud water, cloud ice, and rain and snow precipitation are prognostic variables. Both the large-scale and convective moist processes are parametrised. The convective precipitation is not included in the prognostic precipitation variables, which is not an issue for the IR but requires special treatment in the all-sky microwave, which is strongly sensitive to convective precipitation
(Geer et al., 2018a). However, detrained convective cirrus cloud is represented in the prognostic cloud variables, allowing a good representation of the clouds seen by infrared sounders.

The data assimilation uses an incremental formulation (Courtier et al., 1994) with inner loops run at reduced but increasing resolution, up to TL399 (approx. 50 km). Tangent-linear and adjoint models of cloud and precipitation physics (e.g Lopez and Moreau, 2005) and of other parts of the forecast model, link changes in the control variables (transforms of surface pressure,
horizontal wind vector, temperature, specific humidity and ozone) to changes in dry and moist variables at the observation location. Most available components of the global observing system are assimilated including surface-based platforms (e.g. buoys, surface stations, aircraft, and radiosondes) and satellites (e.g. radiances from infrared and microwave instruments on polar and geostationary platforms, radio-occultation, atmospheric motion vectors and scatterometers for ocean surface wind vectors). This includes all-sky assimilation of microwave humidity sounders and microwave imagers (Geer et al., 2017). These
latter are used to improve the dynamical initial conditions through 'generalised 4D-Var tracing' where the initial conditions are adjusted so that the updated model forecast provides a better fit to the observed patterns of water vapour, cloud and precipitation. It is expected that all-sky IR water vapour sounding channels will improve analyses and forecasts in a similar manner, as they already do in clear-sky conditions (Peubey and McNally, 2009). Full documentation of the ECMWF system is available at http://www.ecmwf.int.

The experiments in this study are run at reduced horizontal resolution: forecasts and data assimilation outer loops are at TCo399, around 25 km; inner loops reach a maximum of TL255, around 80 km. The early delivery assimilation window is dropped, so that forecasts out to 240 h are run from the main 12 h assimilation window. This is the standard configuration for testing new developments at ECMWF, and in most cases its results have been representative of those in the full operational configuration. Experimentation is carried out for two periods of 3 months: from 1st June to 31st August 2017 and from 1st
December 2017 to 28th February 2018; results from the two periods are combined so as to give up to around 360 forecast samples. A control experiment has been run that includes the full observing system but without the 7 IASI water vapour channels, and then a series of experiments (to be introduced later) add these channels with various configurations of observation error and VarQC. Cycle 45r1 of the ECMWF system has been used in most of the work presented here: this is a version that went operational in June 2018.

## 2.2 All-sky infrared assimilation

Full details of the all-sky IR configuration will be given by Geer et al. (2018b) and only an overview is provided here. As mentioned in the introduction, all-sky IR assimilation is first being tested on channels sensitive to upper-tropospheric water vapour and cloud, due to their better linearity and smaller errors (Chevallier et al., 2004). The combination of a forecast model and a cloud-capable observation operator has long been able to make simulations that resemble the real observations



**Table 1.** Details of the 7 mid- and upper-tropospheric water vapour channels to be assimilated in all-sky conditions.

| Channel number | Wavenumber [cm$^{-1}$] | Peak of weighting function [hPa] | Indexed by weighting function |
|---|---|---|---|
| 2889 | 1367.00 | 684 | 1 |
| 2958 | 1384.25 | 662 | 2 |
| 2993 | 1393.00 | 538 | 4 |
| 3002 | 1395.25 | 405 | 7 |
| 3049 | 1407.00 | 604 | 3 |
| 3105 | 1421.00 | 468 | 6 |
| 3110 | 1422.25 | 520 | 5 |

(Chevallier and Kelly, 2002) but still, substantial development has been needed to create an observation operator that has small enough systematic errors for data assimilation. In particular the representation of cirrus cloud has required improvement. In this work, RTTOV v12.2 (Saunders et al., 2018) is used to simulate the IASI observations, using the Chou-scaling cloud scheme with multiple-independent column representation of cloud overlap, and the 'OPAC' water clouds of Matricardi (2005) and the

'Baran' ice cloud of Vidot et al. (2015). Using the ECMWF background as input, this produces monthly mean biases of at worst around $2 - 4\,\mathrm{K}$ in cloudy areas. Compared to the size of observation errors in cloudy areas (see later) this is a negligible bias (see Sec. 3.4). Therefore the observation operator and forecast model are sufficiently accurate in reproducing the observations that all-sky assimilation is a viable possibility.

    ECMWF assimilates (see e.g. Collard and McNally, 2009; Bormann et al., 2016) a subset of 191 out of the 8461 channels

available from IASI, currently from two polar orbiting satellites, Metop-A and Metop-B (Klaes et al., 2007). These channels all have a spectral width of $0.25\,\mathrm{cm}^{-1}$ and provide information on the atmospheric temperature profile and the surface (165 channels at wavenumbers between approximately $649\,\mathrm{cm}^{-1}$ and $875\,\mathrm{cm}^{-1}$), ozone and the surface (16 channels between $1014\,\mathrm{cm}^{-1}$ and $1062\,\mathrm{cm}^{-1}$), mid and upper-tropospheric water vapour (7 channels between $1367\,\mathrm{cm}^{-1}$ and $1422\,\mathrm{cm}^{-1}$) and lower tropospheric moisture (3 channels between $1990\,\mathrm{cm}^{-1}$ and $2015\,\mathrm{cm}^{-1}$). Table 1 gives the details of the 7 mid- and

upper-tropospheric water vapour channels investigated in this work. Note that, although the global mean peak of the weighting function is given, in practice the weighting functions move up and down in the atmosphere by many hundreds of hPa depending on the relative humidity profile of the free troposphere. Figure 10 later gives evidence of sensitivity down to the surface over the Weddel Sea during Antarctic winter. Figures 5a and 6a later illustrate more typical temperature and humidity Jacobians for these channels.

For the operational clear-sky assimilation, a globally constant observation error covariance matrix is used which includes correlations between all the different channels of one observation (Bormann et al., 2016). Observations are thinned to around





100 km spacings to avoid as much as possible the spatial correlations of observation error, which are not modelled. Cloudy scenes are detected and removed using a combination of the McNally and Watts (2003) approach, imager cloud detection (Eresmaa, 2014) and by using a thinning algorithm that always selecting the warmest available pixel based on the observations in a window channel. Aerosol-affected scenes are also detected and removed. A small number of scenes detected as being
completely overcast are assimilated using this diagnosed cloud top as a lower boundary (McNally, 2009) but this does not apply to the WV channels. All selected channels are assimilated over ocean and land, with two main exclusions: sea-ice areas in over $875\,\mathrm{cm}^{-1}$ (which includes the WV channels) and over land, any channel that might be sensitive to the surface. Other quality control techniques include the rejection of channels with normalised departures greater than 2.5, but although many other observation types use variational quality control (VarQC Andersson and Järvinen, 1998) this is not applied to the IASI
observations. Variational bias correction (VarBC) is applied, in common with most other satellite observation types (Auligné et al., 2007) with a globally constant predictor, four airmass predictors based on layer thicknesses across four different ranges, and a third order polynomial in the instrument scan position. The surface skin temperature is treated as a sink variable in observation space, allowing the window channels to update the potentially erroneous first-guess skin temperature.

    Hyperspectral infrared observations are also assimilated from Atmospheric Infrared Sounder (AIRS) and Cross-track In-
frared Sounder (CrIS) using similar configurations to IASI. Further, the mid- and upper-tropospheric water vapour channels from 5 geostationary imagers around the equator. The presence of all this data (plus its equivalent from many microwave sounders) means that changes in forecast quality coming from different usage of IASI water vapour data will not be large, but as will be seen there is still enough sensitivity in the fits of the short-range forecast to other observations that it is possible to clearly measure the impact of the work described here.

In the all-sky IR experiments, it is only the usage of the 7 mid- and upper-tropospheric water vapour sounding channels that has been changed. The error correlations between these channels and the others are set to zero, so that the 7 channels are treated in many respects as an independent instrument. The remaining coupling to the other channels is through the skin-temperature sink variable, and through the thinning that includes a selection of the warmest window channel scene. However this is not expected to have much effect due to the removal, through a screening test, of situations where the 7 chosen channels have
sensitivity to the surface. The principal changes for all-sky assimilation are (i) to stop rejecting cloud-affected observations (but to retain rejection of aerosol-affected scenes); (ii) to use the cloud-capable version of the RTTOV observation operator described above and (iii) to use the situation-dependent all-sky error covariance matrix developed in the current study.

    Some of the more detailed processing is retained from the clear-sky assimilation, such as the 100 km box thinning and the same bias correction model. But most other data selection and quality control processes have been changed to implement all-
sky assimilation. Background quality control is now applied to the whole block of 7 channels, so that either all channels are kept, or all are rejected. This means that the eigenvectors of the observation error covariance matrix remain fixed, allowing the eigenvalues to be scaled in a controlled way as described later. Instead of checking the size of the normalised background departures, it is the size of the normalised eigendepartures (see later) that is checked: if any of the 7 eigendepartures has a magnitude larger than 3, the whole block of WV channels is rejected. Further, the block is rejected if the lowest peaking
channel (channel 2889) has a surface to space transmittance of greater than 0.1. This protects against situations where the



WV channels actually do have significant surface sensitivity, such as on dry days over the Andes. Finally, variational quality control has been activated for the 7 WV channels because it has proved essential to getting good results from all-sky microwave assimilation (Geer and Bauer, 2011). This is a novel development in the context of correlated observation errors, because of the complexities of representing the prior probability of gross error when it is correlated across channels or levels (Ingleby and

Lorenc, 1993). The solution has been to follow the proposal of Andersson and Järvinen (1998) to apply VarQC to the eigen-projected departures, assuming that the prior probability of gross error is independent for each eigenvalue. Further details are given in the appendix. The assumption is that in all-sky assimilation, the gross error modelled by VarQC does not primarily come from radiance-space issues (for example, the failure of an individual channel) but rather from scenes where the analysis struggles to match the observed cloud or precipitation.

## 3    Error covariance matrices

### 3.1    Definitions and concepts

To find the best estimate of the state, $x$, variational assimilation minimises a cost function, presented here in its most simplified form:

$$J\left(x\right) = \frac{1}{2}\left(x - x^{b}\right)^{T}\tilde{\mathbf{B}}^{-1}\left(x - x^{b}\right) + \frac{1}{2}d^{T}\tilde{\mathbf{R}}^{-1}d \tag{1}$$

Here, $x^{b}$ is the background (prior) state and $\tilde{\mathbf{B}}$ its error covariance matrix; this background error determines how far the analysis can go from the background state. As in the introduction the tilde is used to signify the error covariance matrices as applied practically and to distinguish them from the unknown true matrices. $\tilde{\mathbf{R}}$ is the observation error covariance and $d$ are departures between the state and the observations $y$

$$d = y - b - H\left(x\right) \tag{2}$$

where $H()$ is the nonlinear observation operator that maps from state space to observation space and $b$ is a bias correction (here estimated by VarBC). In 4D-Var, the observation operator $H()$ is further extended to include a forecast model that propagates the state from the beginning of the time window (where the analysis is being made) to the time of the observation. For clarity the more complex aspects of the real costfunction used at ECMWF have been ignored: for example modifications are required to estimate VarBC and VarQC parameters as part of the minimisation. Further, the notation has been simplified compared to

that introduced by Ide et al. (1997).

Some of the key concepts of observation error covariance matrices can be understood through the second term on the right hand side, the observation term $J^{O}$. If the observation errors are uncorrelated then the observation error matrix is diagonal and contains the square of the error standard deviations, $\left(\sigma_{i}^{o}\right)^{2}$ for each observation $i$, so that for $N$ observations the $J^{O}$ part of the costfunction can be computed as

$$J^{O}\left(x\right) = \frac{1}{2}d^{T}\tilde{\mathbf{R}}^{-1}d = \frac{1}{2}\sum_{i=1}^{N}\left(\frac{d_{i}}{\sigma_{i}^{o}}\right)^{2} \tag{3}$$





where $d_i$ is the departure computed for each observation. The error of each observation is clearly independent of the others, although the terms in the summation are not independent due to the use of the state to compute $H(\boldsymbol{x})$ in $d_i$. As described in the introduction, the expectation of the background departures is $\mathbf{HBH^T} + \mathbf{R}$, but in the derivation of all-sky observation error models a significant approximation is often made here: if $\mathbf{B}$ were zero (or as discussed earlier, very much smaller than $\mathbf{R}$)

then the background departures normalised by the observation errors, $\frac{d_i^b}{\sigma_i^o}$, should be distributed according to a Gaussian with an expectation of 1. If an observation error model is based directly on the expectation of background departures, as is mostly the case for all-sky assimilation, that model can be validated by showing that it produces a Gaussian PDF of the normalised background departures (Geer and Bauer, 2011).

The gradient of the observation term with respect to the state can also be written as a summation including independent

observation errors,

$$J^O(\boldsymbol{x})' = -\mathbf{H^T}\tilde{\mathbf{R}}^{-1}\boldsymbol{d} = -\sum_{i=1}^{N} \boldsymbol{h_i}\frac{d_i}{(\sigma_i^o)^2} \tag{4}$$

Here $\mathbf{H^T}$ is the transpose of the matrix of partial derivatives of the observation operator $H()$ with respect to the state (the Jacobian) and $\boldsymbol{h_i}$ are its columns, one per observation. Kalnay (2003) explains further the construction of this derivative. In any case the Jacobians of the observation operator are a familiar way to represent the sensitivity of the observations to the state.

If observation errors are correlated, none of the above simplifications can be made because of the off-diagonal terms in $\mathbf{R}$. However, it is possible to project the observations onto an uncorrelated basis using an eigenvector decomposition of the observation error covariance matrix,

$$\tilde{\mathbf{R}} = \mathbf{E\Lambda E^T} \tag{5}$$

where $\mathbf{E}$ is a matrix with its columns being the eigenvectors $\boldsymbol{e_j}$ and $\mathbf{\Lambda}$ a diagonal matrix containing the eigenvalues $\lambda_j$. The

observation costfunction can again be written as a summation in which the errors - now described by the eigenvalues - are independent:

$$J^O(\boldsymbol{x}) = \frac{1}{2}\boldsymbol{d}^T\mathbf{E\Lambda}^{-1}\mathbf{E^T}\boldsymbol{d} = \frac{1}{2}\sum_{j=1}^{N}\left(\frac{\boldsymbol{e_j^T}\boldsymbol{d}}{(\lambda_j)^{0.5}}\right)^2 \tag{6}$$

This is a fundamental change in the way we have to think about observations: when their errors are correlated, we can think about a sum over what we might term the *eigendepartures* $\boldsymbol{e_j^T}\boldsymbol{d}$, with implied observation error standard deviations given by

the square root of the eigenvalues $(\lambda_j)^{0.5}$. If the background error is relatively small, each of the terms in the summation will on average have roughly the same weight in the costfunction. In other words normalised eigendepartures are (excluding the effect of the background errors) all equally important in the costfunction, and hence have equal weight in the data assimilation. Other practical consequences are that first, the eigenvector decomposition can be a useful way of computing the inverse of the $\mathbf{R}$ matrix when computing the costfunction. Second, it gives a practical way to combine a correlated observation error

matrix with variational quality control, which is more easily applied to the now-independent eigendepartures (Andersson and Järvinen, 1998, see appendix).



**Table 2.** Details of the error covariance matrices examined here

| Name |
| --- |
| Operational clear-sky: Bormann et al. (2016) |
| 45r1 clear-sky |
| 45r1 all-sky land/ocean |
| 45r1 all-sky ocean only |
| 43r1 all-sky |

Just as there is an equivalence between the departures and the eigendepartures, there is also what we can call an *eigenjacobian* that gives the sensitivities of each eigenvector to the state. Now, the gradient of the observation costfunction can be computed from this summation:

$$J^O(\boldsymbol{x})' = -\mathbf{H^T}\mathbf{E}\boldsymbol{\Lambda}^{-1}\mathbf{E^T}\boldsymbol{d} = -\sum_{j=1}^{N}\mathbf{H^T}\boldsymbol{e_j}\frac{\boldsymbol{e_j^T}\boldsymbol{d}}{\lambda_j} \tag{7}$$

By analogy to Eq. 4, the eigenjacobian for each eigendeparture is given by $\mathbf{H^T}\boldsymbol{e_j}$. A common problem with the eigendepartures is trying to understand what they respond to physically, so the eigenjacobian gives a useful tool to understand their physical sensitivities.

Finally, these equations have so far been written with one giant observation error covariance matrix, but this is not how current data assimilation systems work. When the assumption is made that observations are uncorrelated in space, but errors are correlated across the channels of one instrument, then the matrix becomes block diagonal. Then the same maths can be applied to the submatrices of $\mathbf{R}$ and subvectors of the departures $\boldsymbol{d}$ for each individual observation as is done in the rest of this study. For further background see Rodgers (2000) and Kalnay (2003).

## 3.2 Clear-sky and all-sky covariance matrices

The all-sky error covariance matrix has been in development for a while, so a number of different versions will be examined here (Tab. 2). The one used in active experimentation is based on background departures from a development version of the all-sky IR assimilation at an earlier cycle (cycle 43r1) using the Metop-A IASI data from a single 12 h analysis window on 3rd May 2016. A number of other matrices have been derived using data from 1 - 20th June 2017 from Metop-A and Metop-B from a passive monitoring experiment using the cycle 45r1 configuration described in Sec. 2 (passive monitoring here means that the background is kept fixed and taken from a control experiment, so it remains unaffected by the change in observation usage). The variety allows an assessment of the robustness of the error estimates. In most cases the sample includes all land and ocean scenes except where orography is greater than 2500 m, and excluding the whole of the Antarctic continent. However the covariances from cycle 45r1 have also been computed on further reduced samples: one keeping clear-skies only, the other





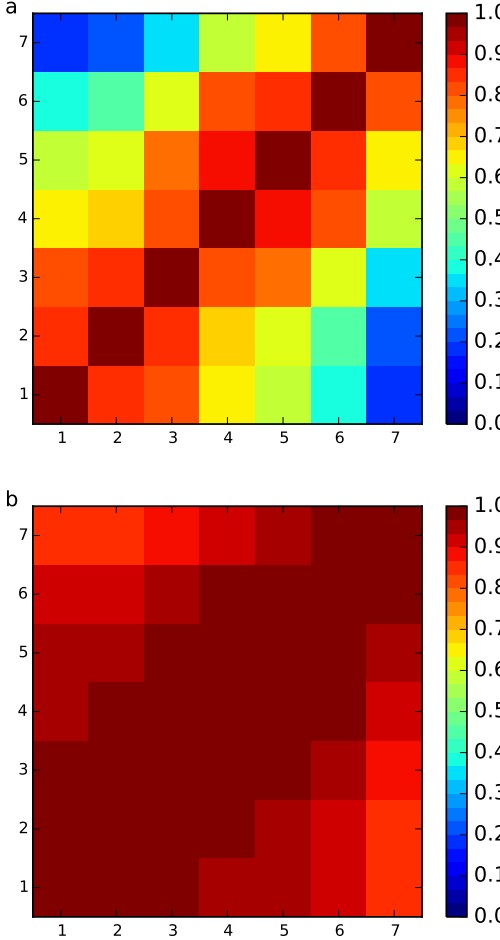

**Figure 1.** Error correlation matrices $\mathbf{C}$ for the 7 selected IASI upper-tropospheric water vapour channels, ordered by ascending altitude of weighting function, from (a) operational clear-sky observation errors; (b) estimated all-sky errors from cycle 43r1 experiments.

all-sky but keeping ocean only. Also, the corresponding part of the operational clear-sky error matrix of Bormann et al. (2016) has been examined.

Figure 1 compares clear-sky and all-sky observation error correlation matrices $\mathbf{C}$ for the 7 IASI upper-tropospheric humidity channels. The error covariance matrix has been decomposed into a correlation matrix and a diagonal matrix of the error standard deviations, $\tilde{\mathbf{R}} = \mathbf{\Sigma}^{0.5} \mathbf{C} \mathbf{\Sigma}^{0.5}$. For display purposes these channels are ordered by the vertical location of their clear-sky weighting functions, from lowest to highest (see Tab. 1). This makes the relationships between channels much clearer than





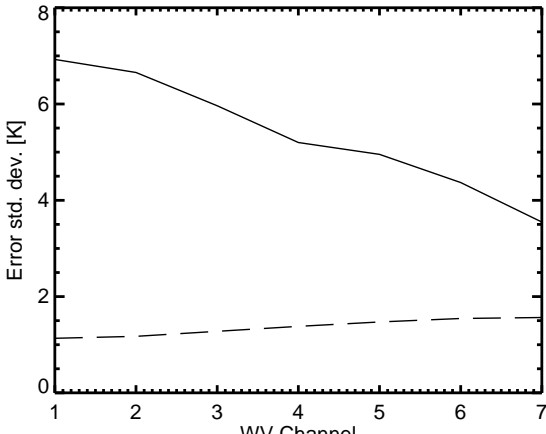

**Figure 2.** Error standard deviations $\mathrm{diag}(\Sigma^{0.5})$ for the 7 selected IASI upper-tropospheric water vapour channels, ordered by ascending altitude of weighting function, from operational clear-sky observation errors (dashed) and estimated all-sky errors from cycle 43r1 experiments (solid)

when following the standard IASI channel ordering, with which interpreting the error covariance matrices becomes difficult (see e.g. Stewart et al., 2014; Bormann et al., 2016). Panel a shows the sub-matrix taken from the observation errors used operationally for assimilation of clear-sky IASI radiances at ECMWF (Bormann et al., 2016). In clear-sky conditions, the lowest peaking channel (IASI channel 2889) and the highest peaking channel (3002) have correlations of only around 0.18, reflecting the minimal overlap in their weighting functions. For channels whose weighting functions overlap more closely, errors become increasingly correlated. This pattern is characteristic of what is thought to be representation error, which is dominant in the water vapour channels (e.g. Bormann and Bauer, 2010; Weston et al., 2014). Panel b shows the correlations of all-sky background departures derived from the 43r1 sample. As expected (Bormann et al., 2011) the presence of cloud in many of the observations leads to much stronger correlations between channels, with minimum correlations of 0.84.

Figure 2 shows the corresponding error standard deviations $\mathrm{diag}(\Sigma^{0.5})$ in clear-sky and all-sky conditions. Error standard deviations are around 1.0 to 1.5 K for operational assimilation. As expected (Geer and Bauer, 2011; Bormann et al., 2011; Okamoto et al., 2014; Harnisch et al., 2016) errors are much larger when cloudy scenes are included, ranging from 3.5 K to 6.9 K. The lower in the atmosphere that the channel observes, the more it is affected by cloud: first, there is a higher chance of the channel seeing cloud; second, the radiative contrast between clear and cloudy skies can be much larger. Hence the highest errors are in the lowest peaking channels.

The eigenvector decomposition $\tilde{\mathbf{R}} = \mathbf{E}\mathbf{\Lambda}\mathbf{E^T}$ provides an alternative view of these error matrices. Figure 3 shows the eigenvectors, ordered by the size of their eigenvalues from highest to lowest (with one exception described below). To assess the robustness of these estimates, the figure includes all the matrices from Tab. 2. At the broadest level, the clear-sky and all-sky eigenvectors are quite similar. Their leading eigenvectors project signals that are similar in all 7 channels, and whether in clear-





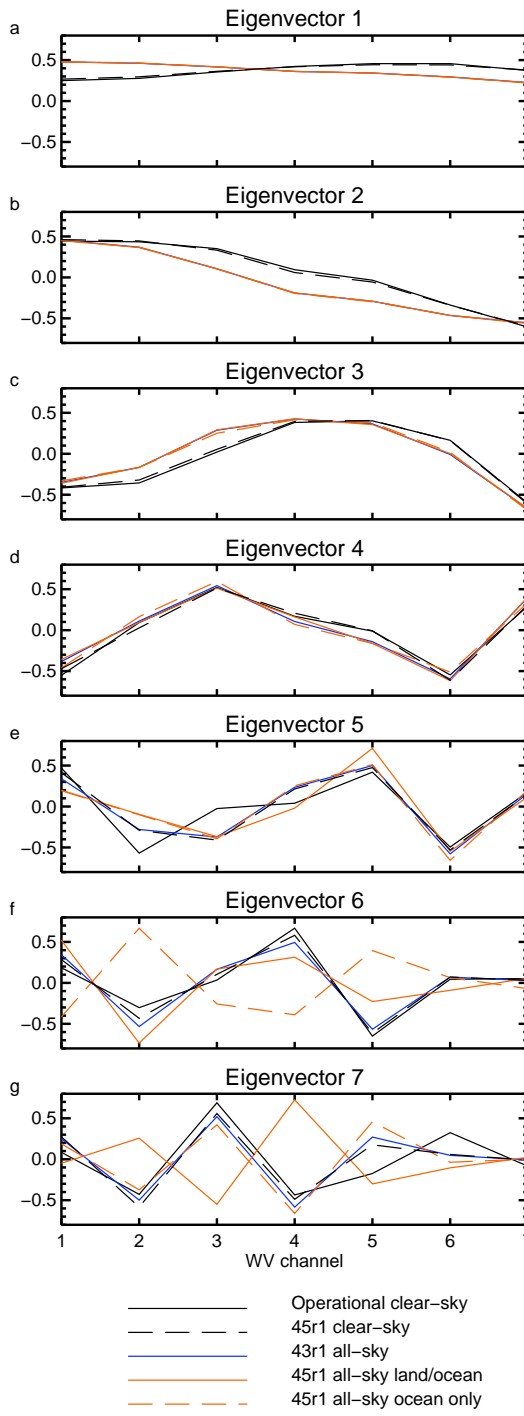

**Figure 3.** Eigenvectors of possible observation error covariance matrices for the 7 assimilated IASI upper-tropospheric water vapour channels. Generally the black lines (clear-sky error matrices) are almost totally overlaid, and similarly the coloured lines (all-sky error matrices) are also often overlaid.





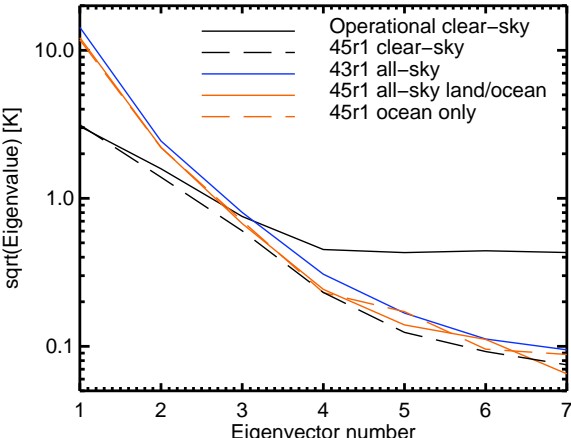

**Figure 4.** Square root of the eigenvalues associated with the eigenvectors in Fig. 3

sky or all-sky, the subsequent eigenvectors represent combinations of channels in a harmonic progression. The apparently large differences in some estimates of the last 2 eigenvectors are simply a matter of opposite signs. The exception to the eigenvalue ordering was made for the operational clear-sky matrix to put the eigenvectors in the same harmonic order: the final three eigenvectors displayed are the 7th, 5th and 6th when ordered strictly by eigenvalue. As will be seen shortly, this ambiguity in

the ordering is probably due to the scaling of the trailing eigenvalues performed by Bormann et al. (2016) which means the last three have almost the same eigenvalue. Perhaps the main difference between clear-sky and all-sky eigenvectors is that relatively higher weight is put on the lower-peaking channels in the all-sky versions; this will be investigated in the next section and is likely due to the additional visibility of cloud (and hence additional variance) in the lower peaking channels. But overall, Fig. 3 emphasises that whatever the source, all the error matrices have eigenvectors with broadly similar structures.

Figure 4 shows the square roots of corresponding eigenvalues ($\lambda_j^{0.5}$). These are the equivalents of the error standard deviations when the errors are correlated (compare Eqs. 3 and 6, and similarly they represent the weight given to each "eigenchannel" in the data assimilation.) The fundamental difference between the clear-sky and all-sky covariance matrices is in the leading two eigenvalues, which are substantially inflated in all-sky conditions. In the clear-sky error matrices, the leading (square-root) eigenvalue is 3.1 K, compared to 12.1 K to 14.3 K in the all-sky versions. The adjustment of the trailing eigenvectors performed

by Bormann et al. (2016) leads to the four trailing (square-root) eigenvalues of the operational clear-sky matrix being all around 0.43 to 0.45 K. The clear-sky and all-sky error matrices derived directly from background departures without any adjustment all have very small trailing eigenvalues. Depending on the estimate, the last (square-root) eigenvalues are around 0.065 to 0.095 K. This represents a strong amplification of information that projects onto the trailing eigenvectors, i.e. amplification of the high-order harmonic combinations of the water-vapour channels. This is not necessarily a desirable feature of an error

covariance matrix. For example both Weston et al. (2014) and Bormann et al. (2016) found they needed to artificially inflate the size of the trailing eigenvalues to improve the conditioning of the 4D-Var solution. Further, Bormann et al. (2015) estimated





trailing eigenvalues that were smaller than those of the instrument error covariance matrix (a hard minimum for observation error) which suggests that some of the diagnosed amplification was unphysical. However, before any adjustments, a strong amplification of the trailing eigenvectors is common to error covariance matrices computed under both clear-sky and all-sky conditions.

In the introduction it has been argued that the Desroziers et al. (2005) observation error covariance diagnostics for clear-sky assimilation may be problematic, and certainly can be bypassed by simpler methods, at least for water vapour channels and particularly for all-sky assimilation. Figures 3 and 4 complete the justification, by comparing the eigenvectors and eigenvalues of the operational clear-sky error covariance matrix to the equivalents from the covariance of clear-sky background departures calculated from the sample of 20 days of 45r1 data. The eigenvectors are essentially identical outside of small differences

in the highest two harmonics. The leading eigenvalues are identical and the trailing eigenvalues differ mainly due to the eigenvalue adjustment performed on the operational matrix. As described by Bormann et al. (2016), the operational matrix was derived from an initial Hollingsworth and Lönnberg (1986) estimate followed by a Desroziers et al. (2005) estimate, and the error variances were inflated by a factor of 1.75 based on detailed tuning experiments. The net result is an observation error covariance matrix that (eigenvalue adjustment aside) is nearly identical to the error covariance of the clear-sky background

departures, at least in the WV channels.

    Figures 3 and 4 also help explore the stability of error covariances estimated from background departures, by looking at covariance matrices computed from three different samples of data. The first four eigenvalues are nearly identical in all three estimates computed from all-sky departures (although slightly different from the clear-sky estimates). The trailing eigenvectors show a little more variability, although most of it is simply due to changes in sign. That the trailing eigenvectors seem less

stable might also be an indication that they should not be amplified as much as the raw error covariance matrices might suggest. A larger difference is between the eigenvalues of the 43r1 and 45r1 estimates, with the 43r1 versions being larger by around 20% (comparing the square roots). This is likely due to the worse quality of the all-sky radiative transfer model used at cycle 43r1 and because the older cycle had generally larger short-range humidity errors associated due to an older formulation of background error covariance matrix. To avoid the expense of re-running costly data assimilation experiments, it has been

preferred to stick with the original cycle 43r1 covariances that were roughly estimated several years ago at the start of this work. However, apart from the 20% overestimate of the size of the errors, the 43r1 estimates are clearly robust and little would be gained by using the later versions.

## 3.3 Jacobians and eigenjacobians

The sensitivity of the observed radiances to geophysical quantities is described by the Jacobian matrix of partial differentials

of the observation operator. For each channel $i$ this can be calculated using the adjoint of the observation operator applied to a vector $\mathbf{1}_i$ that contains a 1 at position $i$ and zeroes elsewhere, i.e. $\mathbf{H^T 1}_i$. In one sense this is just a way of extracting the columns of the adjoint observation operator matrix, $\mathbf{h}_i$, but the equivalence with the eigenjacobian $\mathbf{H^T e}_j$ (Sec. 3.1) is complete if we think of the vectors $\mathbf{1}_i$ as the chosen eigenvector basis when the error matrix is diagonal. Figure 5a shows the temperature Jacobians of the seven IASI upper-tropospheric water vapour channels, computed for a midlatitude clear-sky





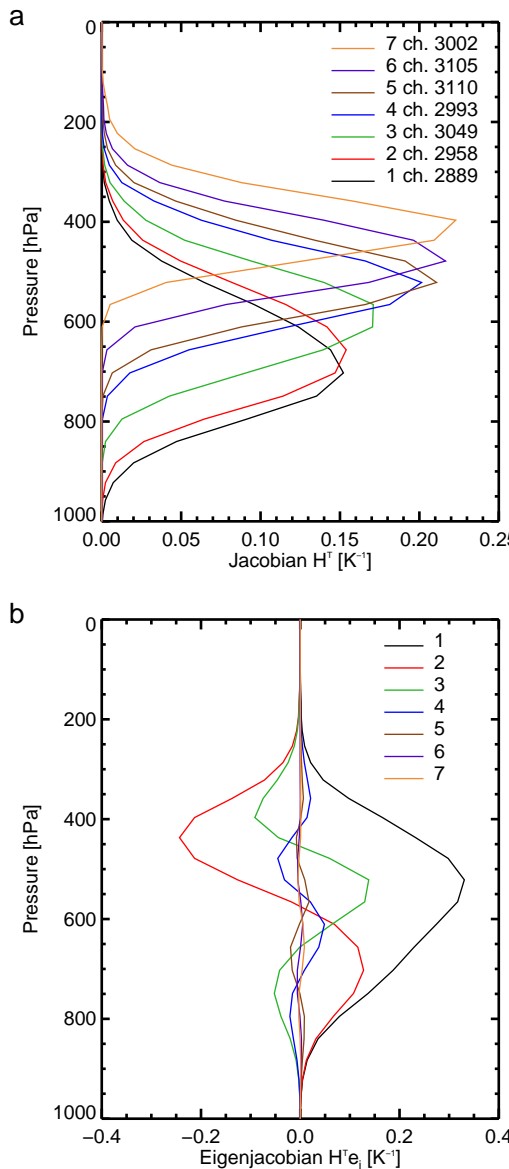

**Figure 5.** Temperature parts of channel Jacobians ($h_i$) and eigenjacobians ($\mathbf{H^T}e_j$) for the all-sky 43r1 observation error covariance matrices, for a clear-sky profile



profile (this profile is exactly the one supplied with RTTOV that is used in the standard example to compute the Jacobian matrix). For this profile, the temperature Jacobians span levels from around 800 hPa to around 300 hPa. However, since this sensitivity comes mainly from water vapour absorption, in conditions of high free-tropospheric humidity these Jacobians could be pushed up higher in the atmosphere; in lower humidity all the Jacobians may sit close to the top of the humid boundary

layer. The eigenjacobians are shown in panel b. As with the eigenvectors themselves (Fig. 3) the eigenjacobians are a series of harmonic functions representing oscillating features of increasingly fine vertical scales. For example, the leading eigenvector is sensitive to a weighted average temperature across much of the troposphere, whereas the fourth eigenvector is sensitive to vertical temperature oscillations with a 'wavelength' of around 250 hPa. The eigenjacobians of the trailing eigenvectors become increasingly small, in other words the higher-order harmonic combinations of channels have increasingly little sensitivity to

atmospheric temperature changes, based as they are on the differences between neighbouring channels. As will be illustrated later, it is the smallness of the trailing eigenvalues that amplifies this sensitivity.

    Figure 6 shows the corresponding Jacobians and eigenjacobians for the (specific) humidity sensitivity. These show broad sensitivity into the stratosphere (the tropopause in this profile is at around 290 hPa) that are usually thought to have little practical effect due to the very small humidity amounts at these levels (and in the ECMWF system, because humidity increments are

suppressed above the tropopause). Once again the eigenjacobians are a series of harmonic functions in the vertical, although with very little direct sensitivity in the trailing eigenfunctions.

    The role of the eigenvalues of the error covariance matrix in amplifying sensitivities to higher-order vertical oscillations can be seen in Fig. 7, which shows the temperature sensitivities of the costfunction for each eigendeparture, extracted from Eq. 6 as $\mathbf{H^T} e_j/(\lambda_j)^{0.5}$ or equivalently the columns of $\mathbf{H^T E \Lambda^{0.5}}$. In contrast to the eigenjacobians (Fig. 5b) the additional weighting

provided by the square root of the eigenvalues causes the broad vertical structures to be downweighted whereas the fine-scale vertical sensitivities have been amplified. Many others have already illustrated this effect of modelling observation error correlations, for example by means of single observation test cases (Bormann et al., 2011), by 1D-Var retrievals (Weston et al., 2014), and by examination of the eigenvalues or the error correlations (Bormann et al., 2016). However, the eigenjacobian and the costfunction sensitivity of each eigenchannel, projecting into geophysical space, are a very straightforward way of

diagnosing these effects.

    The effect of cloud is explored in Fig. 8. To the previously clear-sky profile, an artificial single-layer water cloud has been added at 700 hPa, filling the whole satellite field of view (i.e. cloud fraction is 1.0). The effect of a nearly opaque cloud like this is to truncate the Jacobians at the cloud top, and to provide much stronger temperature sensitivities at this level. The effect on the eigenjacobians is similar, but with a less-strong increase in sensitivities at the cloud top.

It is also possible to compare the eigenjacobians of the clear-sky and all-sky error covariance matrices, as shown in Fig. 9. It has already been noted that the all-sky eigenvectors themselves seem to give a little more weight to the lower-peaking channels, at least for the first 4 eigenvectors. The effect is to move the eigenjacobian sensitivities down in the atmosphere by around 50 hPa, but it does not fundamentally change the sensitivity of each eigenvector. Along with the evidence in the previous section, it seems that the all-sky error covariance matrix is not fundamentally different from the clear-sky matrix,

except for the much higher eigenvalues on the leading (and to some extent, second) eigenvector. The fundamental shape of the



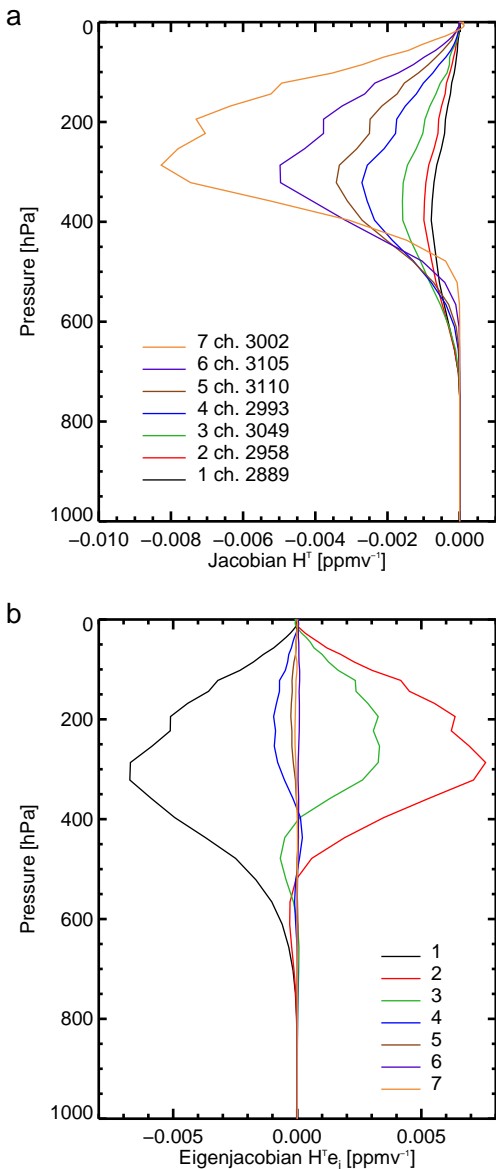

**Figure 6.** Humidity parts of channel jacobians ($h_i$) and eigenjacobians ($\mathbf{H}^{\mathbf{T}} e_j$) for the all-sky 43r1 observation error covariance matrices, for a clear-sky profile



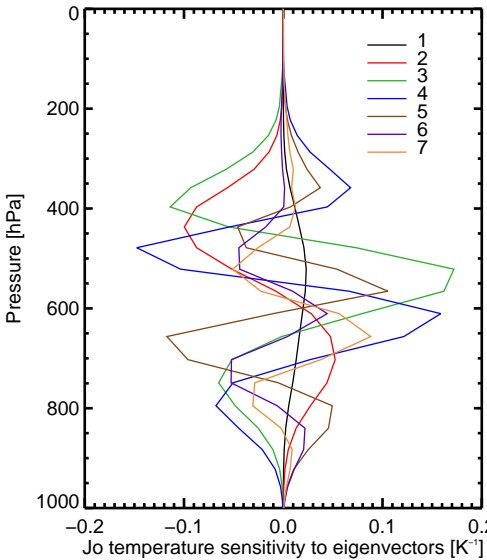

**Figure 7.** Temperature sensitivity of the Jo costfunction, by eigenvector ($\mathbf{H^T E \Lambda^{0.5}}$), showing (by comparison to Fig. 5b) the role of the eigenvalues of the error covariance matrix in amplifying sensitivities to smaller vertical wavelengths, and reducing sensitivities to broad vertical features.

error covariance matrix seems to be determined by the clear-sky sensitivities of the different channels (cf the raw Jacobians) and is not much altered when clouds are included.

### 3.4 Behaviour of eigendepartures

When assimilating data with a diagonal observation error representation, it is the background departures that are routinely analysed for Gaussianity (desirable) and bias (undesirable). For example Fig. 10 shows the mean normalised background departures in the highest and lowest peaking of the seven WV channels, $d_i/\sigma_i^o$, using observation errors from the diagonal of the 43r1 observation error matrix. There are biases of up to at least 0.5 times the observation error in the tropics and subtropics suggesting both an underprediction of deep convection over land areas (e.g. Africa, India, Argentina) and a slight excess over tropical oceans (e.g. Atlantic) that is consistent with results from all-sky microwave radiances (e.g. Geer and Baordo, 2014). A smaller negative bias is prevalent in some ocean areas, and there is a distinct bias over the ice in the Weddel sea in the lowest peaking channel. To assimilate this data, the sea-ice areas are removed, and observation error inflation in cloudy areas will help reduce the impact of cloud-related biases. However, even with regional biases even as large as 0.5 to 1 times the observation error standard deviation, the assimilation of all-sky microwave radiances at ECMWF has shown benefits (e.g. Lonitz and Geer, 2017).

When using a correlated observation error representation, it also makes sense to explore the Gaussianity and bias of the eigendepartures. Figure 11 shows the 20-day mean of normalised background eigendepartures $\boldsymbol{e}_j^T \boldsymbol{d}/(\lambda_j)^{0.5}$ from the 45r1





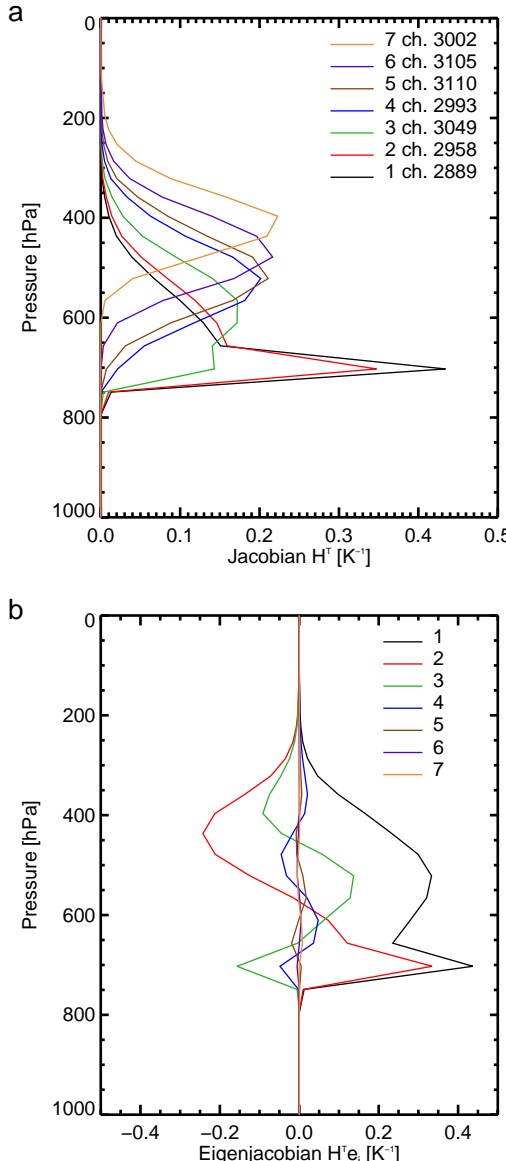

**Figure 8.** Temperature parts of channel jacobians ($h_i$) and eigenjacobians ($\mathbf{H^T}e_j$) for the all-sky 43r1 observation error covariance matrices, with a full-coverage single-layer maritime cumulus water cloud of $10^{-3}$ kg/kg inserted at 700 hPa





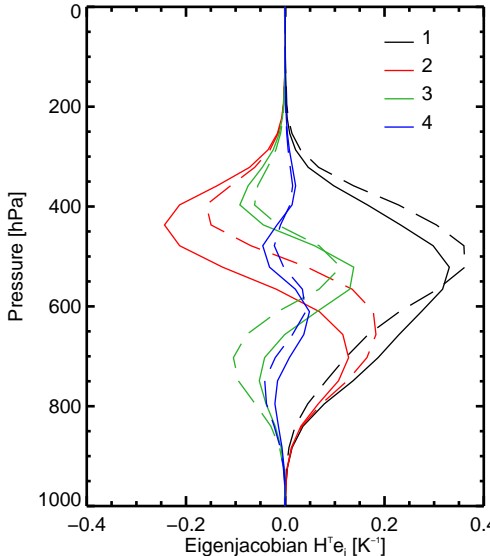

**Figure 9.** Temperature parts of eigenjacobians ($\mathbf{H}^{\mathbf{T}}e_j$) for the first 4 eigenvectors of either the operational clear-sky (dashed) or all-sky 43r1 (solid) observation error covariance matrices.

passive monitoring experiments described in the previous section. Biases in some of the trailing eigendepartures are quite large, requiring an extended colour scale for this figure compared to Fig. 10. Eigenvector 1 has bias patterns similar to those in the ordinary departures, though with reduced amplitude. The trailing eigenvectors appear to have amplified bias patterns that are not obvious from the brightness temperatures. For example there is very little bias in the Atlantic at 20°S in Fig. 10, but eigenvectors 4, 5 and 6 have biases peaking at around -1, +1 and +1.5 respectively. It is likely that some subtle inter-channel biases (whether coming from the observations or the modelling) has been amplified here, but tracking it down would be difficult. A normalised bias of 1 in eigenvector 6 could have been generated by a systematic bias between channels 2 and 4 of around 0.01 K (inferred from Figs. 3 and 4). Clearly correlated observation errors demand extremely low inter-channel bias.

The mean eigendepartures in Fig. 11 use the VarBC bias correction and the background from an experiment with the full cycle 45r1 configuration (including clear-sky use of the 7 IASI water vapour channels). An interesting question is whether VarBC can help reduce eigendeparture bias if allowed to evolve in an experiment in which the observations are assimilated actively with the new error covariance matrix. Bormann et al. (2015) showed that VarBC bias corrections can adapt when observation error correlations are activated, responding to the way that error correlations modify the weights given to different observations. Hence the equivalents of Fig. 11 have also been computed from the baseline all-sky assimilation experiment with correlated errors (described in full later). The global biases between eigenchannels are indeed reduced (not shown) but still all of the more regional, larger amplitude patterns remain. It is hard to say whether the biases have been reduced by VarBC or by changes in the analysis and short range forecast, but later Fig. 18 shows vertically oscillating mean changes in temperature between the experiments and the control that are consistent with Fig. 11, suggesting that the eigendeparture biases





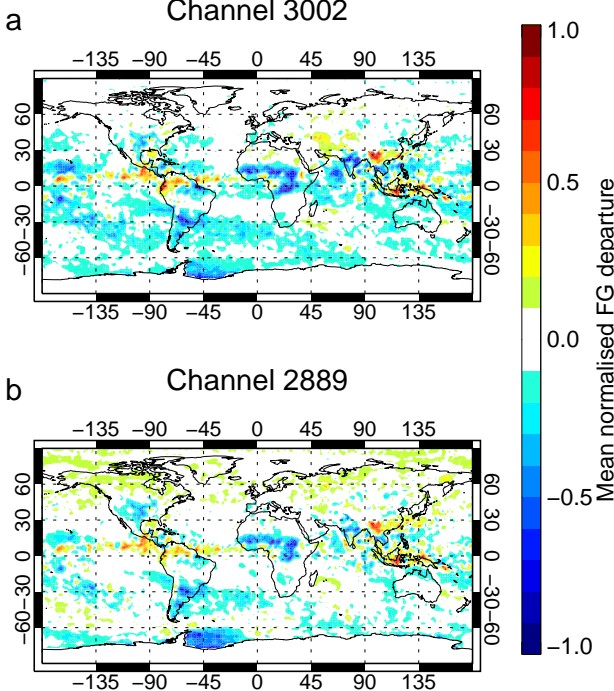

**Figure 10.** Mean normalised background departure, $d_i/\sigma_i^o$, after bias correction, in IASI channels 3002 and 2889, the highest and lowest peaking of the 7 WV channels. Based on a sample of passive background departures from 1-20 June, 2017, with screening only for orography over 2500m and for the Antarctic continent.

have not been completely corrected by VarBC but have caused a mean shift in the analysis and short-range forecast. Another reason why VarBC might not correct at least the regional biases is that the airmass bias predictors are not correlated with the predominantly localised bias patterns. If these patterns come from cloud-related biases in the model or observation operator, it would still be very difficult to find predictors for them (see e.g. Lonitz and Geer, 2017). The interaction of VarBC and error

5   covariances deserves further study, but it is still clear that very low inter-channel biases are required to safely use interchannel error correlations. This may be achievable with a well-calibrated instrument in clear-sky conditions, but when model error is prevalent, such as in all-sky assimilation, this could be a problem.

Making the assumption that background errors are small relative to observation errors, an appropriate observation error model should generate normalised background eigendepartures with a PDF similar to a standard Gaussian. Figure 12 shows

10   the distribution of eigendepartures using the 43r1 error covariance matrix. For eigenvectors 2 to 7, Gaussianity is achieved within the range of +/- 3. Although there are excessively heavy tails outside this range, these will be removed by quality control in the data assimilation and this will only lose a small amount of data. For eigenvector 1, there is much too sharp a peak and too broad tails, the typical picture seen for all-sky departures if an adaptive error model is not used (e.g Geer



**Figure 11.** Mean mean normalised background eigendepartures, $e_j^T d / (\lambda_j)^{0.5}$, after bias correction. Based on a sample of passive background departures from 1-20 June, 2017, as Fig. 10





**Figure 12.** Probability density functions of normalised background eigendepartures (thin line) and a standard Gaussian (dashed line). For eigenvector 1, normalised background departures with all-sky error scaling makes the sample more Gaussian (thick line). Based on the sample of departures from 1-20 June, 2017, as Fig. 10.




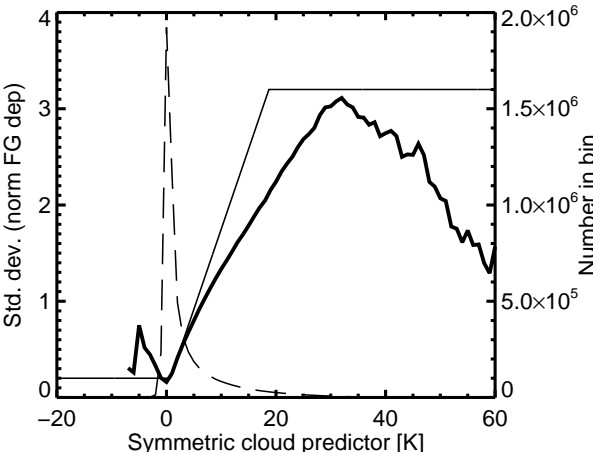

**Figure 13.** Standard deviation of normalised background eigendeparture of eigenvector 1, computed in 1 K bins of the cloud proxy variable $C$ (thick), an approximate piecewise linear fit $s_1$ (thin), and the number of observations per bin (dashed).

and Bauer, 2011). This suggests that most of the signal of cloud displacements goes into the 1st eigenvector. Comparing the first eigenvalues of the clear-sky and all-sky covariances matrices in Fig. 4 suggests that the eigenvalue should be smaller in clear-skies, giving more weight to the clear-sky data, and larger in cloudy skies, reducing the weight and bringing down the amount of observations going into the tails of the PDF. Hence the error model can be improved by scaling the 1st eigenvalue

as a function of a cloud proxy variable, following the standard approach used in all-sky assimilation (Geer and Bauer, 2011). The second eigenvector is also not perfectly Gaussian, and likely it too projects some of the cloud signal, based on its larger eigenvalue compared to clear-sky conditions (Fig. 4). However for the moment, this work leaves it unscaled.

The best choice of cloud proxy variable for all-sky IR assimilation is still a matter of research. Okamoto et al. (2014) and Okamoto (2017) suggested using the difference between all-sky and simulated clear-sky brightness temperatures; this was

used successfully to inflate cloudy observation error in the assimilation of all-sky Himawari data in a tropical cyclone test case (Honda et al., 2018). Alternative suggestions from Harnisch et al. (2016) and Minamide and Zhang (2017) have been considered, but were not as successful (see Geer et al., 2018b, for more information). The predictor of Okamoto et al. (2014) gave good results when used to drive the adaptive error scaling model that will be introduced shortly, producing the PDF given by the thick line in Fig. 12a. This shows good Gaussian behaviour within the range +/-3 and though there is an excessive warm

tail (corresponding to situations where more cloud is modelled than observed) this will be removed by quality control.

Figure 13 shows how the error model was derived, following the normal approach. The normalised background eigendepartures were binned as a function of the symmetric cloud proxy variable computed from the lowest peaking channel 2889:

$$C = \frac{1}{2}\left(H_{\mathrm{CLR}}^{2889}(\boldsymbol{x_b}) - y^{2889}\right) + \frac{1}{2}\left(H_{\mathrm{CLR}}^{2889}(\boldsymbol{x_b}) - H^{2889}(\boldsymbol{x_b})\right) \tag{8}$$





Here superscript 2889 signifies the part of the observation vector or observation operator corresponding to the chosen channel and $H_{\text{CLR}}$ is the nonlinear observation operator used in clear-sky mode (i.e. ignoring any presence of cloud in the background profile). The cloud proxy variable is known as 'symmetric' because it is a mean of the same quantity (cloud effect, i.e. cloudy minus clear brightness temperature) across both observations and background, which helps avoid sampling biases in the as-

similation. Note that the formulation used by Okamoto et al. (2014) takes the absolute value of the observed and background cloud effect before computing the mean cloud effect. In the current work negative values of cloud effect are retained, but this is is only a minor difference, because as seen in Fig. 13 the population of negative symmetric cloud amounts is tiny.

Standard deviations were computed from each population, and then this distribution could be approximately fitted by the piecewise linear function:

$$s_1 = \min\left(\max\left(\frac{C+0.5}{6.0}, 0.2\right), 3.2\right). \tag{9}$$

The standard deviations do not have strictly linear behaviour and they reduce towards high symmetric cloud amounts, but as in other similar work the piecewise linear fit is allowed to produce excessively large values in this region. These will correspond to an over-cautious downweighting of heavily cloudy scenes. However, for the vast majority of the population with symmetric cloud amounts between 0 and 5, the fit is quite good.

To generate a situation-dependent error covariance matrix, $s_1$ can be used as a scaling factor to adjust the leading eigenvalue of the error covariance matrix so that eigendepartures are now calculated as $e_j^T d / s_j (\lambda_j)^{0.5}$ where $s_1$ for eigenvalue 1 is the scaling function just calculated and $s_j$ for $j > 1$ is 1, i.e. no scaling is applied to other eigenvalues. The scaling factor bottoms out at 0.2 in clear-sky conditions, boosting the weight given the these observations compared to what they would get with a fixed covariance matrix. The scaling factor reaches a maximum of 3.2 in very cloudy conditions, significantly reducing the

weight given to observations that are likely to be affected by mislocation error. If $\mathbf{S}$ is a matrix containing $(s_j)^2$ on the diagonal and zeroes elsewhere, then this creates an adaptive observation error matrix that can be computed as

$$\tilde{\mathbf{R}} = \mathbf{E}\mathbf{S}^{0.5}\boldsymbol{\Lambda}\mathbf{S}^{0.5}\mathbf{E}^{\mathbf{T}} \equiv \mathbf{E}\mathbf{S}\boldsymbol{\Lambda}\mathbf{E}^{\mathbf{T}} \tag{10}$$

Figures 14 and 15 show respectively the error standard deviation and correlation of this new matrix, for fully clear and fully cloudy skies ($s_1 = 0.2$ and $s_1 = 3.2$). The clear sky errors are close to the existing operational clear-sky errors, both in terms

of correlation and standard deviation. This would be expected given the similarity of clear-sky and all-sky eigenvectors (Fig. 3) and the primary difference between clear-sky and all-sky eigenvalues being in the leading eigenvalue (Fig. 4). In fully cloudy skies, this error model produces even stronger correlation between channels, and the error standard deviations are significantly boosted in all channels. This is consistent with error covariance matrices computed directly from the highly cloudy sample (not shown). Hence, situation-dependent eigenvalue scaling seems to be a viable technique to create an adaptive error covariance

matrix for all-sky assimilation.





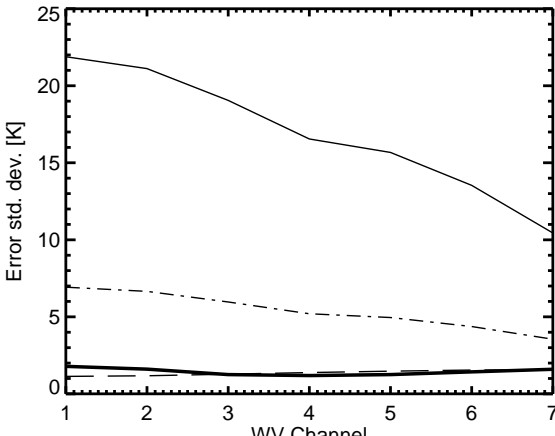

**Figure 14.** Error standard deviations for the 7 selected IASI upper-tropospheric water vapour channels, ordered by ascending altitude of weighting function, using the adaptive all-sky error covariance matrix with scaling factors $s_1 =$3.2 (thin), 1.0 (dot-dashed) and 0.2 (thick), compared to those from the operational clear-sky matrix (dashed).

## 4 Results in all-sky data assimilation

### 4.1 Finding the best configuration

The scaled observation error covariance matrix was tested in a total of 6 months of full-cycling data assimilation experiments using the ECMWF all-sky IR assimilation framework at cycle 45r1 that is summarised in Sec. 2 and described in more detail by

Geer et al. (2018b). The control excludes the 7 IASI upper-tropospheric water vapour channels from the clear-sky assimilation framework, but other IASI channels are assimilated, along with the rest of the global observing system used operationally at ECMWF. The first experiment, labelled 'all-sky baseline', tests the initially proposed configuration with the 43r1 all-sky error covariance matrix with eigenvalue scaling according to cloud amount, as well as VarQC applied to the 7 WV eigenchannels. A number of other experiments, to be introduced later, test different configurations of the error covariance matrix, as well as the

necessity for VarQC.

Initial results are shown in Fig. 16 using the standard deviation of departures from assimilated Advanced Technology Microwave Sounder (ATMS) observations, at analysis (left) and background (right). All results have been normalised by the standard deviations from the control experiment. Since the ATMS observations are used in the analysis they are not an independent reference in the left hand panel. As such, adding a new observation could conceivably draw the analysis fit away from

ATMS (and increase the standard deviations) if it brought new information with significantly lower errors. However, the ATMS fit to the background is a valid measure of the quality of the 12 h forecast. Unfortunately the 'all-sky baseline' experiment using the proposed error covariance matrix gave significantly worse results than previously seen with the all-sky IR framework at ECMWF (Migliorini et al., 2014; Geer et al., 2018a). Fits to the ATMS temperature channels (6–15) were degraded, particu-





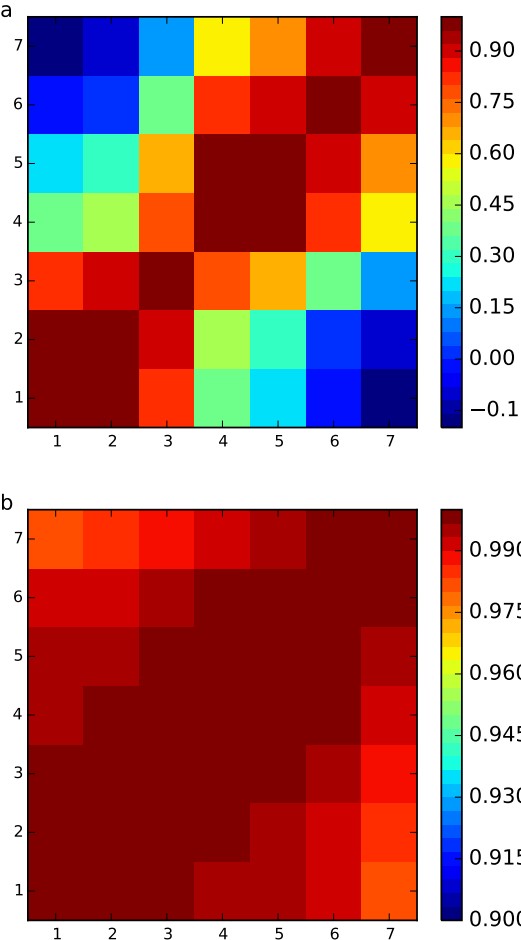

**Figure 15.** Error correlation matrices for the 7 selected IASI upper-tropospheric water vapour channels, ordered by ascending altitude of weighting function, showing the form of adaptive error covariance matrices for (a) clear-sky and (b) full cloud conditions. Note the substantial difference in the colour scale between the two plots.



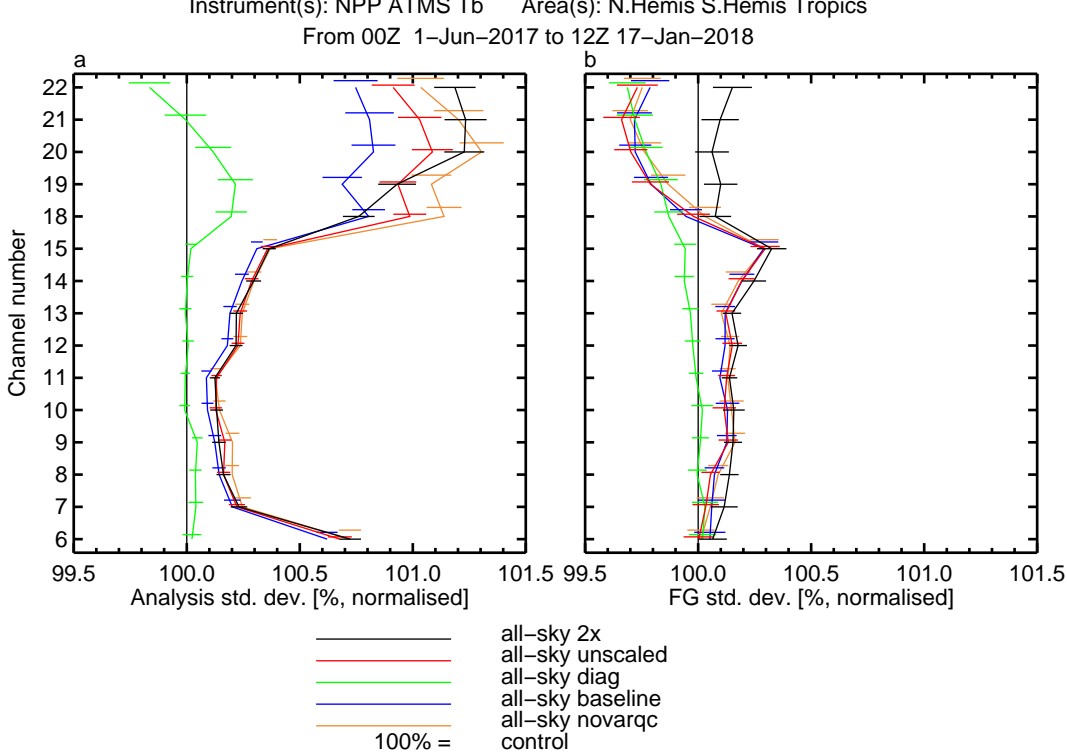

**Figure 16.** Standard deviations of (a) analysis and (b) background departures from the global set of assimilated ATMS observations, normalised by the standard deviations of the control and presented as a percentage. The error bars give the 95% confidence range for differences compared to the control, based on a Student's t-test.

larly in the analysis and particularly for the higher channels. Channel 15 is sensitive to temperatures in the upper-stratosphere, around 2 hPa, so it is strange that tropospheric cloud and water vapour information should affect these levels. However, all-sky microwave assimilation is thought to generate or modify gravity or equatorial wave activity which propagates into the stratosphere in the ECMWF system. This has sometimes appeared beneficial (e.g Geer et al., 2014) and more recently as the weight

5    of all-sky data in the system has grown, it has started to appear problematic (e.g. Lean et al., 2017). However the stratospheric degradation caused by all-sky IR assimilation is much worse than previously seen with all-sky microwave observations. Further, fits to the ATMS mid-upper tropospheric humidity channels (18–22) are degraded substantially in the analysis, and there is very little improvement in the background.

Other experiments in Fig. 16 explore the impact of configuration choices related to observation errors in the all-sky assimila-

10   tion. First, 'all-sky novarqc' is as 'all-sky baseline', but VarQC has been deactivated for the 7 WV channels. This demonstrates that VarQC has been beneficial to the quality of the water vapour analysis fit to ATMS, but is unlikely on its own to be a solution to the problems. Second, 'all-sky unscaled' shows the effect of using a constant, rather than adaptive, version of the error covariance matrix (in other words, $s_1$ is always 1). Again there are degradations in the WV analysis fits compared to the initial




configuration, indicating that adaptive error scaling also has some benefit. However there does not appear to be any benefit to forecasts over the non-adaptive matrix. Third, 'all-sky 2x' follows the approach of Bormann et al. (2016) by inflating the error standard deviations unilaterally by an additional factor 2 in the hope that the problematic behaviour is reduced. The additional inflation does not make any difference to the analysis or background fits in the temperature channels, but it does degrade the

fit in the humidity channels, resulting in a framework where the all-sky IASI WV assimilation makes no improvements to the forecast at all, only degradations. In this respect the lack of impact on the background is consistent with the results of 3.5 times scaling from Bormann et al. (2016): the basic all-sky matrix is already around 1.75 times larger than the clear-sky Desroziers et al. (2005) estimates, so the 'all-sky 2x' experiment would be equivalent to a 3.5 times inflation in their terms. Hence unilateral inflation has reduced the beneficial impacts while not addressing the underlying problem.

A final test 'all-sky diag' explores whether error correlations are necessary at all, here by using the diagonal of the all-sky error covariance matrix (clearly this requires the adaptive error scaling to be switched off). This simple approach results in much better analysis fits to ATMS, but it generates similarly underwhelming improvements in the background fits. The initial conclusion is that something in the error covariance matrix must be causing problems in the quality of the analysis.

   Weston et al. (2014) found substantial problems with the conditioning of their clear-sky IASI error covariance matrix so

they had to increase the value of all eigenvectors with an additive inflation that increased the trailing eigenvectors by nearly two orders of magnitude. Bormann et al. (2016) found only minor problems with conditioning (note this was in the context of the matrix that was already inflated 1.75 times, and the second-level preconditioning used at ECMWF should make the minimisation less susceptible - see later) but they made further adjustments to the trailing eigenvalues. These adjustments had a particularly big effect on the trailing eigenvalues of the submatrix 7 WV channels as is clear from Fig. 4. Further, Sec. 3 has

highlighted a number of other potential concerns with the level of amplification provided by the trailing eigenvalues. Hence, a second set of experiments explored the adjustment of the trailing eigenvalues so they were no smaller than either 1 or 0.37 (the latter chosen fairly arbitrarily as being close in size to the fourth eigenvalue, see Fig. 4). The adaptive scaling of the first eigenvalue remains unaffected, but in the adjustment to 1.0, for example, eigenvalues from 3 to 7 all have to be inflated to that level. Just as scaling the leading eigenvalue affects the error standard deviation (Fig. 14), adjusting the trailing eigenvalues

must also have an effect. However, it is much smaller. For the adjustment to 1.0, the observation error standard deviations are increased by no more than around 0.1% in the fully cloudy case. In the fully clear case, the increase was more significant at between 7% and 22% depending on the channel, so it likely does slightly weaken the observational constraint in clear sky conditions.

   Figure 17 shows that adjustment of the trailing eigenvalues puts an immediate stop to the problem of degradations in the

ATMS temperature channels, while providing much better fits to the water vapour channels than is achieved with the diagonal all-sky error model. All-sky IASI assimilation now improves the analysed and background fits to ATMS water vapour channels by around 0.6%, which is comparable to the current impact of the 7 WV channels in the operational clear-sky approach (not shown). Just replicating the clear-sky results may not seem much, but in the context of all-sky assimilation this counts as a success. Although both versions of the adjustment provide good results, adjustment to 0.37 has a small advantage in the fit to

the ATMS observations at background. The advantage is larger and has statistical significance in the ATMS humidity channels,



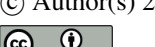

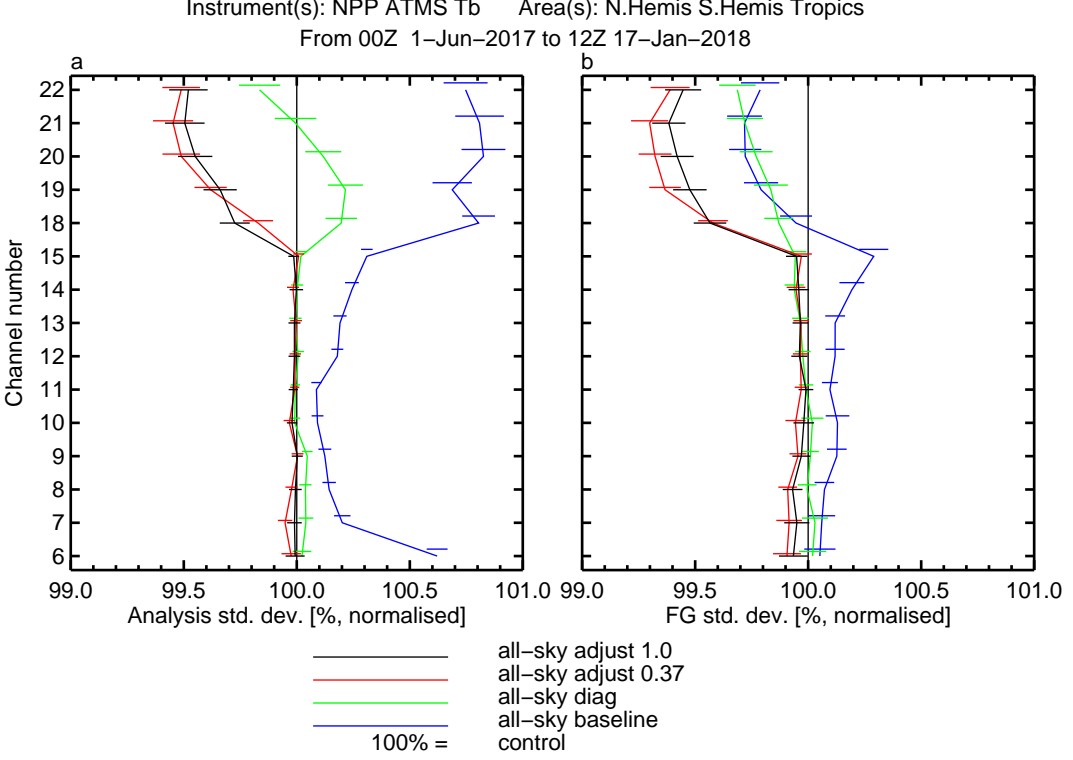

**Figure 17.** As Fig. 16 but for a set of experiments that inflate some of the trailing eigenvalues.

and smaller and less significant in the temperature channels. Fits to other observations (such as other microwave humidity and temperature sounders, radio-occultation, and radiosonde, not shown) confirm this picture. However forecast scores based on verification against the experiment's own analysis seem to favour the 1.00 adjustment over 0.37 but in the early forecast range such statistics are very difficult to interpret (not shown). More results from the all-sky IASI assimilation framework, and comparisons to clear-sky assimilation, are reserved for the study of Geer et al. (2018b). For now, the conclusion is that a suitable all-sky observation error model must represent observation error correlations, and it must use VarQC and all-sky error inflation of the leading eigenvector, but most importantly of all, the trailing eigenvalues must be adjusted. The next section explores why the adjustment is so important.

## 4.2 Problems with error correlation models

It has been shown that using the raw correlated error model, all-sky IASI WV assimilation degrades the analysis. Particularly strange is the degradation in stratospheric and lower tropospheric temperatures, when the IASI WV observations are mainly sensitive to the mid and upper-troposphere. A first possible issue, explored in depth by Weston et al. (2014), would be the conditioning of the observation error matrix. In the Met Office system they describe, the 4D-Var costfunction is pre-conditioned using the background error matrix, meaning that the conditioning of the 4D-Var solution is largely determined by that of





**Table 3.** Statistics of the quality of the minimisation from the last inner-loop iteration of 4D-Var: median condition number and mean number of iterations, across all data assimilation cycles in the experiments.

| Experiment | Condition number | Number of iterations |
|---|---|---|
| Control | 2420.5 | 30.2 |
| All-sky baseline | 3600.9 | 34.1 |
| All-sky novarqc | 4122.4 | 35.2 |
| All-sky diag | 2472.9 | 30.4 |
| All-sky unscaled | 3749.7 | 34.0 |
| All-sky 2x | 3914.9 | 34.5 |
| All-sky adjusted 1.0 | 2429.3 | 30.4 |
| All-sky adjusted 0.37 | 2432.1 | 30.5 |

the observation error covariance matrix. Poor conditioning increases the number of iterations required for convergence. The minimisation at ECMWF uses a similar background error preconditioning but also a second level preconditioner based on the leading eigenvectors of the Hessian of the costfunction (Fisher and Andersson, 2001) which means it should be less sensitive to these problems.

Table 3 shows the condition numbers and number of iterations required in the final minimisation of the incremental 4D-Var in the current experiments. The condition number is given as the median across all cycles in the experiment, to reduce the influence of occasional high condition numbers that can occur infrequently in all experiments (due to occasional instabilities in the stratosphere of the TL and adjoint forecast model, it is speculated). Condition numbers using the diagonal error matrix, or either of the two error matrices with adjustments to the trailing eigenvalues, are around 2420 – 2480, comparable with the

condition number in the control. In these cases, the minimisation takes around 30 iterations to converge. In contrast, the error matrices which do not adjust the trailing eigenvalues produce condition numbers in the range 3600 – 4100 and take around 34 – 35 iterations to converge. This is quite different from the results of Bormann et al. (2015) where giving full weight to the trailing eigenvctors only added a maximum 2 iterations, rather than 5 here, suggesting that all-sky assimilation makes the conditioning problem worse. The condition numbers also give further information on some of the sensitivity tests. Compared

to the baseline all-sky error matrix (4D-Var condition number 3600) turning off either VarQC and leading-eigenvector scaling makes this worse (going to 4122 and 3750 respectively) and a unilateral 2x scaling also makes the condition number worse (going to 3914). The additional 4 iterations in the all-sky baseline experiment could increase the cost of data assimilation by around 10% if extrapolated across all minimisations.

Poor conditioning cannot directly explain the degradation in the quality of the analysis and forecast, unless the minimisa-
tion stops before full convergence has been achieved. This is not thought likely as the hard iteration limit is 50 in the current experiments, so even at 35 iterations the minimisation has stopped due to satisfying the standard convergence criterion. Hence





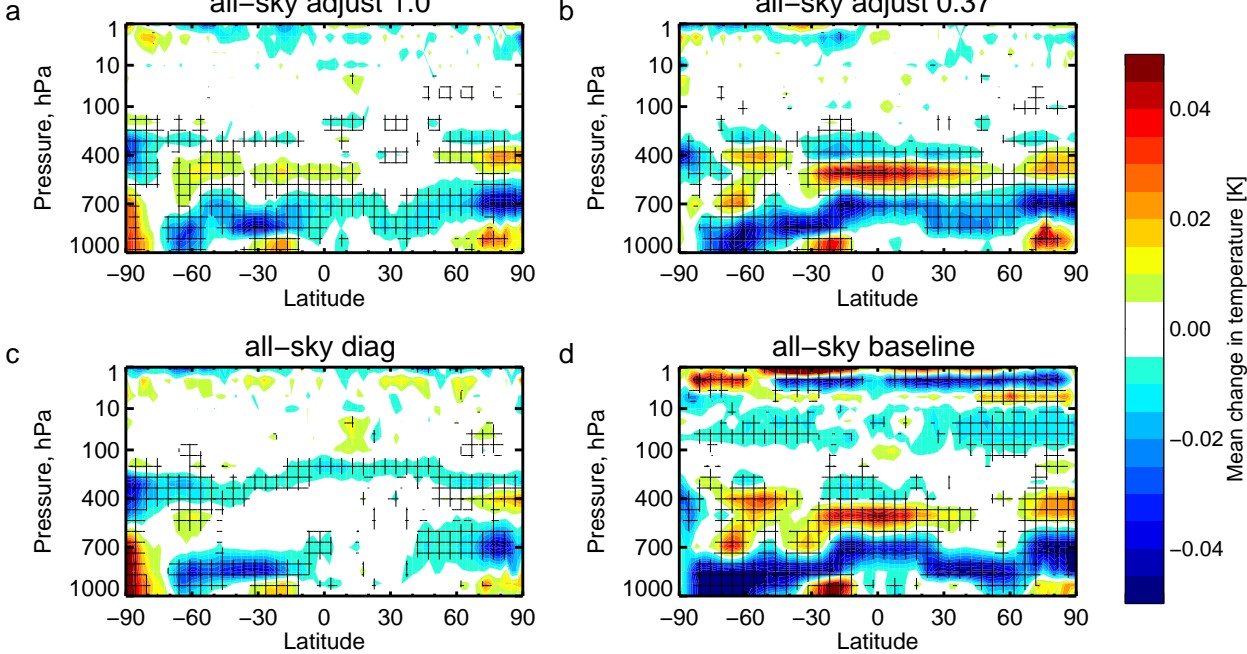

**Figure 18.** Mean change in zonal mean temperature analysis between experiments and the control. Cross-hatching indicates 95% confidence with Sidak correction for 20 independent tests.

something more than just the conditioning is required to explain the degradations coming from the non-adjusted error covariance matrices.

Section 3.4 has highlighted a number of other potentially problematic aspects of the trailing eigenvectors and the very small eigenvalues that amplify their sensitivity. The amplification of bias is one possibility. Figure 18 shows the change in zonal mean temperature in four of the experiments. A common feature of the experiments using correlated observation errors is up to a 0.1 K cooling somewhere around 700 hPa to 950 hPa, with a smaller warming around 500 hPa to 700 hPa and sometimes additional 'ripples' above and below. Adjusting the trailing eigenvectors reduces the amplitude of this bias pattern so that it is similar to that in the experiment with diagonal errors. This suggests that subtle interchannel biases have indeed been amplified by the trailing eigenvectors.

Figure 11 showed that eigenvectors 4, 5 and 6 have areas of quite strong bias. In areas of the southern subtropics eigenvector 4 predominantly has a positive bias and 5 and 6 a negative bias. Figure 7 shows that eigenvectors 4, 5 and 6 all have a strong lobe of sensitivity between around 750 hPa and 900 hPa, with eigenvector 4 having a negative sensitivity and 5 and 6 a positive sensitivity. Therefore the biases in all three eigenvectors often act in the same direction, which gives the data assimilation a possibility to correct the biases by changing the temperature profile. Figure 19 shows the temperature increments that would be required to correct a bias pattern of -0.5 bias in eigenvector 4, +0.5 bias in eigenvectors 5 and 6, and zero bias in the other eigenvectors, similar to the dominant zonal pattern at 15°S. The increments can be computed very approximately by ignoring





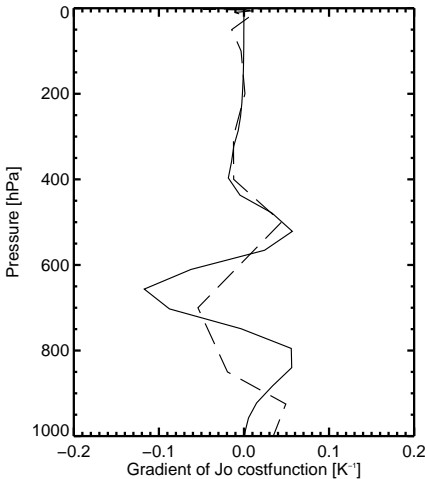

**Figure 19.** Temperature increment (solid line) required to correct normalised background eigendepartures of [0,0,0,-0.5,0.5,0.5,0] using the example Jacobians from Fig. 5, examining the effect of the dominant bias patterns at 15°S in Fig. 11. Overlaid (dashed line) are the mean changes from Fig. 18d at 15°S – these have only been computed on standard pressure levels, hence the jaggedness in the vertical

the background term in the 4D-Var costfunction and solving Eq. 7 for $(J_O)' = 0$. (Precisely these approximate increments can be computed from the eigenjacobians and normalised eigendepartures as $\sum_j \mathbf{H^T} \boldsymbol{e_j} \frac{e_j^T \boldsymbol{d}}{\lambda_j}$.) This suggests there should be an 0.5 K warming at 800 hPa and below, a 0.1 K cooling at 600 hPa to 700 hPa, and further warming and cooling changes above. This is partly consistent with the actual bias changes in Fig. 18d at 15°S, which have been overlaid. However perfect agreement

could hardly be expected here: in practice, the temperature Jacobians of the 7 water vapour channels are not static but instead they vary widely depending on the WV profile. Further, in the 4D-Var analysis the size and location of the increments is of course also controlled by the background errors. However, this is a clear demonstration that the trailing eigenvectors can amplify subtle interchannel biases (particularly seen in eigenvectors 4,5 and 6 in this case) in a way that would give a vertically oscillating pattern of changes to the mean analysed temperature. These mean changes could have caused the degradations in

fit to ATMS temperature observations, particularly the large degradation in the lowest peaking channel, ATMS 6, which has maximum sensitivity between 500 hPa and the surface.

An equally worrying aspect of Fig. 18 is that correlated error models cause mean changes in the temperature of the upper stratosphere, another region where ATMS temperature channels suggest a degradation in the analysis (channels 12 – 15, see Fig. 17). Again, adjustment of the trailing eigenvalues is able to remove this effect. However, a mechanism is required to

15 propagate information (whether correct or not) from the mid/upper-troposphere to the upper-stratosphere. One mechanism could be gravity waves. It is clear from Figs. 5 and 7 that the trailing eigenvalues amplify information that contains high vertical resolution oscillatory temperature patterns. These oscillatory patterns could directly generate gravity waves in the analysis, and these gravity waves could propagate temperature and wind changes across the depth of the atmosphere within one 12 h analysis cycle. Maps of the additional increments generated by adding the 7 WV channels are shown in Fig. 20, which confirms that





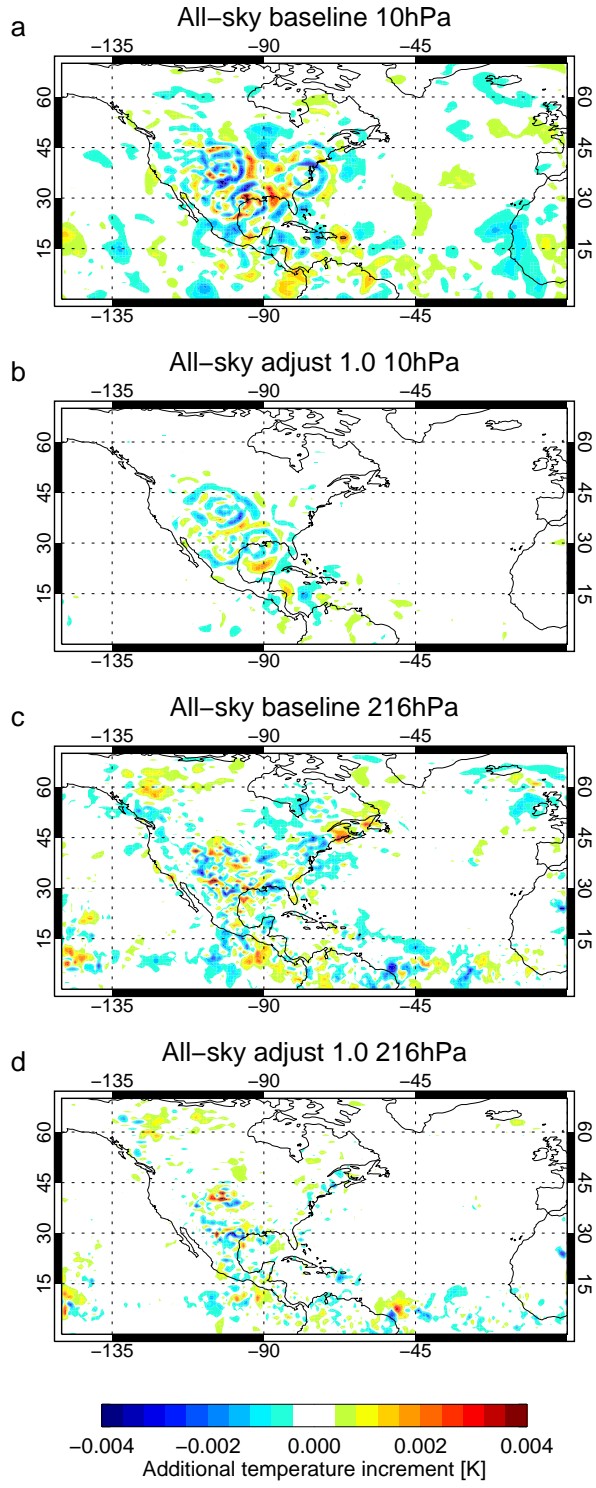

**Figure 20.** Additional temperature increments generated by all-sky IR assimilation on top of the otherwise full observing system, at 00Z on 1st June 2017 at the beginning of the experiments where the background is identical across all experiments.



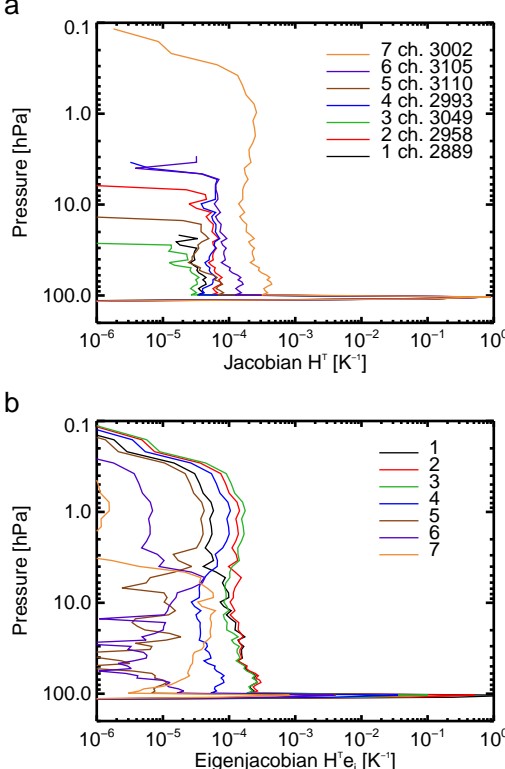

**Figure 21.** Temperature parts of the Jacobians $h_i$ and eigenjacobians ($\mathbf{H^T}e_j$) for a tropical profile with an optically thick, full-coverage cloud placed exactly at the temperature tropopause.

the baseline all-sky IR assimilation does indeed generate substantially more increments in both the upper-troposphere and stratosphere, and that these increments seem to occur in similar locations from the stratosphere to the mid-troposphere. At 10hPa, the increments have patterns resembling gravity waves (the circular structures over north America) and to a lesser degree equatorial waves, although the wave nature of the stratospheric increments is clearer from global figures (not shown).

5    For the baseline all-sky experiment, the increments clearly degrade the analysis, as measured by fits to ATMS stratospheric temperature channels (Fig. 17) and a much stronger 2% degradation in analysis fit to radio-occultation measurements (not shown). The full all-sky error covariance matrix generates much greater wave intensity than the adjusted or (not shown) diagonal versions. Hence gravity waves must have a role in explaining the degraded analysis and early-range forecast.

A final possibility that could generate increments in the stratosphere would be if, under certain conditions, the eigenjacobians

10    could directly generate sensitivities in the stratosphere. The physics of the situation can potentially generate this: for example over a cold tropical cloud top at around 185 K, emission from the stratosphere can increase brightness temperatures by several Kelvin (see e.g. Fritz and Laszlo, 1993). A small emission from the stratosphere could perhaps be sufficiently amplified by the trailing eigenvectors that it could strongly affect the analysis. None of the standard Jacobians or eigenjacobians explored in



**Table 4.** Relative size, in percent, of stratospheric temperature sensitivity in a tropical profile with and without optically thick cloud at the tropopause (sum of temperature Jacobian in stratosphere of total sum of temperature Jacobian). Results given either for the regular Jacobian ('TB') or for eigenjacobians ('Eigen').

| Chan. / Eigen. | Clr TB | Clr Eigen | Cld TB | Cld Eigen |
|---|---|---|---|---|
| 1 | 0.00 | 0.01 | 0.08 | 0.25 |
| 2 | 0.00 | 0.04 | 0.20 | 1.42 |
| 3 | 0.00 | 0.10 | 0.05 | 6.20 |
| 4 | 0.04 | 0.10 | 0.20 | 7.04 |
| 5 | 0.01 | 0.06 | 0.132 | 13.1 |
| 6 | 0.01 | 0.59 | 0.367 | 10.3 |
| 7 | 0.11 | 0.53 | 1.23 | 59.5 |

Figs. 5, 7 or 8 show any evidence of stratospheric temperature sensitivity. Note that stratospheric humidity sensitivities can be ruled out in this case because humidity increments are zeroed above the tropopause in the ECMWF system. Figure 21 is based on an example tropical profile with an artificial, full coverage, optically thick cloud placed exactly at the tropopause. Here as expected there is some sensitivity to the stratosphere. In brightness temperatures this is seen most strongly in the highest peaking channel 3002, as might be expected since water vapour absorption is strongest in this channel. However the sensitivity is insignificant compared to the sensitivity to temperature at the cloud top. In contrast eigenvectors have varying sensitivity to the temperature at the cloud top, with the trailing eigenvectors having least of all. Although the stratospheric sensitivity remains small, it becomes an increasing proportion of the total. Table 4 shows the relative fraction of the temperature Jacobian or eigenjacobian that is in the stratosphere, for the tropical profile with and without cloud. At the most extreme, eigenvector 7 has more temperature sensitivity to the stratosphere than it does to the temperature cloud top.

Figure 21 and Tab. 4 show that there is a real possibility for trailing eigenvectors to generate temperature increments in the stratosphere, though the effect in the data assimilation system ultimately depends on how the relative impact of background errors, observation errors (i.e. eigenvalues) and the relative sensitivities to water vapour, cloud fraction and cloud amount. To test whether direct stratospheric sensitivities are important, an experiment was run with the temperature sensitivities of the tangent-linear and adjoint observation operator set to zero above 100 hPa for the all-sky IASI assimilation. This did not significantly affect the results, suggesting that direct sensitivity to the stratosphere is not a significant issue. It would be much harder to set up experiments to test the relative influence of other possible mechanisms, i.e. biases and gravity waves, so this must be left for future investigation.



## 5 Conclusion

This study describes the first observation error covariance matrix to have both inter-channel error correlations and an adaptive scaling. It has been developed to support the all-sky assimilation of 7 IASI infrared water vapour sounding channels at ECMWF that will be described in more detail by Geer et al. (2018b). In common with most other observation error models used for all-

sky assimilation, it uses an inflation factor that is a piecewise linear function of a symmetric cloud proxy variable (Geer and Bauer, 2011, symmetric meaning the variable is a mean of both the observed and simulated cloud amount). The cloud proxy represents the influence of cloud on the brightness temperatures, here based on the difference between all-sky and simulated clear-sky brightness temperatures following Okamoto et al. (2014), but choosing the lowest peaking channel to represent all 7 water vapour channels.

The error covariance model starts by computing the covariance of a global sample of all-sky background departures. This follows standard practice in all-sky assimilation where the observation errors are assumed to be much larger than the background errors, due to large errors of representation, so that the covariance of the background departures is a reasonable approximation to the observation error itself. To make the error covariance model adaptive, the leading eigenvalue is scaled as a function of the symmetric cloud proxy variable. This takes advantage of the fact that most of the cloud signal projects onto the leading

eigenvector, which is true for the 7 water vapour channels examined here but would not necessarily be true for more diverse sets of channels.

Another novelty of this work is the application of VarQC alongside a correlated observation error matrix (mostly described in the appendix). This follows the proposal of Andersson and Järvinen (1998) of applying VarQC to the eigendepartures. Knowing how to apply VarQC alongside interchannel observation error correlations will make it feasible to extend these techniques to

all-sky microwave, where VarQC is certainly necessary to making successful forecasts.

Experiments have been run over a total of 6 months with a control that uses the full observing system minus the 7 IASI WV channels. The focus has not been on the quality of the all-sky IR assimilation itself (which will be reported in the other study) but on the different possible configurations of the error covariance model and other associated settings. The results have been summarised based on analysis and background fits to ATMS temperature and humidity sounding channels, and some

diagnostics of the assimilation system. However they are backed up by many other observation fits, not shown here.

The baseline all-sky assimilation has a problem that it degrades analysis fits to ATMS in both humidity channels (sensitive to mid and upper troposphere) and temperature channels sensitive to troposphere and stratosphere. Although in the background forecast it improves humidity fits (indicating that it does provide some benefits) degradations in the temperatures persist into the forecast, particularly in the stratosphere. These degradations appear to come from additional gravity waves and tropical

waves created by the all-sky assimilation. Compared to the baseline configuration:

- Turning off the adaptive scaling makes the analysis fits worse and increases the condition number of 4D-Var. This is likely due to the under-weighting clear-sky observations and overweighting cloudy observations. Hence the adaptive scaling is worthwhile, but there is no evidence that it improves the quality of forecasts compared to a globally constant matrix. A possible explanation is that all 7 eigenvectors of the error covariance matrix contribute equally to the observation




costfunction and thus (filtering due to background errors aside) have equal weights in the assimilation. So although the scaling makes a big difference to the error covariances in brightness temperatures, it only affects a small part of the observations' influence on the assimilation.

- Using diagonal errors (without adaptive scaling) prevents the degradations in the analysis and in the temperature forecast, and provides similar benefit to humidity forecasts. This showed that the problems in the baseline configuration must come from the correlated part of the observation error model.

- Applying an additional unilateral inflation to the error variances was not helpful in reducing and rather it reduced the weight of the observations so far that all-sky assimilation provided no benefit at all to the forecast.

- Variational quality control (VarQC) is also useful to avoid further degrading analysis fits. However, it has little effect on the subsequent forecast which is in contrast to earlier results from microwave data (e.g Geer and Bauer, 2011). This is likely because because cloud in IR water vapour channels has more linear effects than does precipitation in microwave imager channels. This is also evident from the PDFs of background eigendepartures which are well behaved and Gaussian, with very few outliers after all-sky scaling and after quality control. Hence, VarQC may not be that necessary in the case of all-sky IR water vapour channel assimilation.

The problems with the baseline error covariance model were found to come from the trailing eigenvectors. Adjusting these eigenvectors, i.e. not allowing eigenvalues to be smaller than a fixed number (either 1 or 0.37 in the two experiments run here) got rid of the analysis and forecast degradations across the temperature and moisture fields and across stratosphere and troposphere. In these experiments with adjusted trailing eigenvalues in error covariance matrix, all-sky assimilation gave the best results of all, with a 0.6% improvement in background fits to ATMS humidity channels.

Another step beyond previous work has been the introduction of the eigendeparture and eigenjacobian as useful diagnostics of a data assimilation system that uses inter-channel correlated observation errors. The eigenjacobians show the sensitivity of the eigendepartures to the state, indicating that the leading eigenvector responds to a broad vertical average of temperature and moisture, and trailing eigenvectors see increasingly higher harmonics of this, but with increasingly little direct sensitivity because they are computed from the differences between channels. However, when weighted by the square of the eigenvalues (the equivalent of observation error when using a diagonal error matrix) the sensitivities to the vertically broad features are reduced and to the high-periodic vertical features are enhanced. Along with examination of the monthly mean eigendepartures, this helps establish a much clearer picture of how correlated error covariance matrices change the sensitivities of the data assimilation system, complementing earlier work (e.g. Weston et al., 2014; Bormann et al., 2016). As interchannel observation error covariance matrices become more widely used, it is likely that examination of eigendepartures and eigenjacobians should be a regular part of monitoring the observing system and of establishing the sensitivity of new instruments. It may become increasingly necessary to start applying VarQC, quality control and observation monitoring to the eigendepartures instead of (or in addition to) the raw brightness temperatures as done in the current study. However, this may not be as easy for wider selections of channels because if the channel basis changes (for example if some channels are unavailable or discarded for



good physical reasons) the eigenvalues and vectors will also change. However, the development of all-sky and all-surface data assimilation should make it increasingly unnecessary to discard certain channels, making it easier to retain a full channel set in all situations.

To explain the problems encountered with the trailing eigenvectors, this study has described four potential complications with representing interchannel observation error correlations. Already well-known is the issue of ill-conditioning. As explained by Weston et al. (2014) the use of $\mathbf{B}$ preconditioning means that 4D-Var minimisation algorithms can be sensitive to ill-conditioning in the observation error matrix. Including interchannel observation error correlations results in a substantial increase in the condition number of the observation error matrix, and this can also drive an increase in the condition number of the Hessian of the 4D-Var costfunction, causing slower convergence. In the current experiments, and despite a more sophisticated two-level preconditioning, the baseline correlated observation errors increase the condition number of the final minimisation of 4D-Var by around 50% and require 15% more iterations to converge. This has been solved (as in Bormann et al., 2016) by inflating the trailing eigenvalues of the error covariance matrix, which improves the conditioning so that the minimisation is just as efficient when all-sky IR assimilation is activated. However, since the minimisation has still iterated to convergence, poor conditioning cannot explain the worse temperature forecasts, and in particular worse fits to other observations in the stratosphere, that are associated with the generation of additional gravity and tropical wave activity

Three new potential issues with error covariance matrices have been outlined that could help explain the worse short-range temperature forecasts:

1. **Amplification of bias:** Very small interchannel biases can project onto the trailing eigenvectors and be strongly amplified by the very small trailing eigenvalues. For example a 0.01 K bias between two water vapour channels could generate a normalised weight of 1 in the costfunction, which is significant. The correlated error representation amplifies biases that are mostly invisible in the brightness temperatures themselves. When the IASI WV channels were assimilated with the baseline covariance matrix, there were temperature changes in the analysis of around 0.1 K with a vertically oscillating pattern. This pattern was linked to regional patterns of bias in eigenvectors 4, 5 and 6.

2. **Generation of gravity waves:** The trailing eigenjacobians of the IASI WV channels have vertically oscillating temperature sensitivities that resemble gravity waves, with vertical wavelengths of a few hundred hPa (e.g. a few km). The assimilation of all-sky IASI observations increases gravity wave and equatorial wave activity in the stratosphere, with a consequent degradation in fits to other observations sensitive to stratospheric temperature and wind.

3. **Amplification of unwanted sensitivities:** The combination of a global all-sky error covariance matrix, and a relatively extreme situation like high cloud in a tropical profile, leads to the generation of unexpectedly high sensitivities to the stratosphere in the trailing eigenvectors. However, based on an experiment that zeroed the temperature sensitivities to the all-sky IASI observations in the stratosphere, this was not a significant cause of the problems in the current experiments.

Whether the problem was a amplification of bias, or gravity waves, it could be addressed by adjustments to the trailing eigenvalues that are also useful for improving conditioning. The most likely hypothesis is the amplified sensitivity to gravity waves,





which is backed up by the observed large increase in the prevalence of stratospheric gravity waves and tropical waves in the increments generated by the all-sky assimilation with the baseline error covariance matrix. However, as with eigenvalue adjustment purely for conditioning purposes, the solution has come at the cost of losing any useful information that might project onto the trailing eigenvectors. Adjustment to 0.37 had slightly better results than to 1.0, suggesting that over-inflating

the trailing eigenvalues probably does lose useful information. None of the potential issues appear to be problems with the error covariance matrices themselves, but rather with the ability of the current data assimilation framework to make use of the full spectrum of information contained in the observations.

All-sky radiance assimilation is a particularly difficult case for a correlated observation error model: first, inaccuracies in cloud modelling mean biases are common; second, assimilating data across all both clear and cloudy states makes for highly

heterogeneous (heteroskedastic) error covariances. A solution might be to compute error covariance matrices for a number of different sub-populations. For example a series of error covariance matrices could be computed from departures that have been binned according to a cloud proxy variable. This approach is being tested for the all-sky microwave assimilation at ECMWF. There is also scope for improving the initial observation error model described here: perhaps the second eigenvector needs cloud scaling too; maybe there could be a more detailed exploration of error scaling and different methods for diagnosing

observation error (as Bormann and Bauer, 2010; Bormann et al., 2016); finally it would be good to repeat the sensitivity tests (e.g. the importance of VarQC and cloud scaling) on the final adjusted matrix. However the error matrix has provided good enough results that all-sky IR assimilation can now match or surpass the clear-sky assimilation of the equivalent channels (Geer et al., 2018b) which is a starting point for further development.

It is also possible that clear-sky assimilation using correlated observation errors could be similarly affected by the amplifica-

tion of sensitivities to bias, gravity waves, and unexpected parts of the atmosphere that current assimilation can struggle to deal with. Problems may not have been apparent so far due to the routine inflation and adjustment of eigenvectors (e.g. Weston et al., 2014; Bormann et al., 2016). Hence there is likely to be wider applicability to these results than just to all-sky assimilation. As the representation of observation error correlations becomes more common, this may require a more fundamental shift in the practice of data assimilation than is currently recognised. The eigenjacobians and eigendepartures may become a standard

part of observation monitoring and preparations for new instruments. As mentioned above, inflation of trailing eigenvectors is convenient but it probably discards valuable information with high vertical resolution. It might be necessary to accept increased costs of minimisation to use some of this information, or alternatively to further improve preconditioning techniques. It might also be necessary to work out a better way of assimilating information on gravity waves.

## Appendix A: VarQC and correlated errors

VarQC has never been applied to operational hyperspectral IR assimilation at ECMWF, so Bormann et al. (2016) did not have to worry about it when implementing their error covariance matrix. The full theory for applying VarQC in the presence of correlated observation error (Ingleby and Lorenc, 1993; Andersson and Järvinen, 1998) is complex and potentially expensive, involving the inverse of all $2^n$ observation error covariance matrices that cover all possible selections from $n$ observations (for





example $n$ is 7 in this work). Andersson and Järvinen proposed several ways to simplify the problem. The most appealing in the current context is projection onto the eigenvectors of the error covariance matrix in order to diagonalise the problem. VarQC models gross observation error as having a prior probability of $A$. This error is represented by a flat (or rather top-hat) distribution that extends across the range of allowable values of the background departure $d_i$, i.e those that would not be

rejected by the routine QC check on the size of the normalised background departure. This range $\pm l$ is defined as a multiple of the normalised background departure $d_i/\sigma_i^o$ although when used in practice (as here) its size is often different from the actual background departure rejection threshold.

If VarQC is applied to the eigendepartures, it means the model for gross error is also flat across a range of eigendepartures, rather than of brightness temperature departures. Traditional kinds of gross error, such as a bad measurement in a single satel-

lite channel, will project onto all eigenvectors and hence an independent top-hat error model in eigendepartures is incorrect. However, VarQC is not used in all-sky assimilation for dealing with traditional gross error so much as to downweight situations where the analysis struggles to match the observations, particularly those associated with cloud and precipitation errors (see Geer and Bauer, 2011). In other words VarQC is being used to deal with non-Gaussian representation error. In the current work, cloud 'misplacement' errors seem to mostly project on the leading eigenvector, and if gravity waves can also be considered

a type of representation error, they would project mainly on the trailing eigenvectors. Hence applying VarQC to the eigende-partures could actually improve the modelling of non-Gaussian representation error by properly accounting for the way it is correlated in the brightness temperatures.

To apply VarQC for IASI, settings of $A = 0.5$ and $l = 5$ have been used, the same as for all-sky microwave assimilation. The diagonalised costfunction and gradient (Eqs. 6 and 7) would thus be the obvious way to apply VarQC. However in the ECMWF

system it has been easier to do the diagonalisation using the eigenvector decomposition of the error correlation matrix, i.e.

$$\tilde{\mathbf{R}} = \mathbf{\Sigma}^{0.5}\mathbf{C}\mathbf{\Sigma}^{0.5} = \mathbf{\Sigma}^{0.5}\mathbf{E_c}\mathbf{\Lambda_c}\mathbf{E_c^T}\mathbf{\Sigma}^{0.5} \tag{A1}$$

This is because both the costfunction (Eq. 3) and its gradient (Eq. 4) are in practice at ECMWF both computed from the intermediate product $\mathbf{C}^{-1}\mathbf{\Sigma}^{-0.5}\mathbf{d}$ that is saved for later reuse in the gradient computation, which is recovered by applying $\mathbf{H^T}\mathbf{\Sigma}^{-0.5}$. The eigenvector decomposition of the error correlation matrix is different to that of the covariance matrix, but

likely the same arguments apply regarding representation error. The VarQC implementation for correlated error can then be inferred by rewriting Eqs. 6 and 7 using this alternative eigenvector basis to provide the diagonalised '$J_o^N$' and '$\nabla J_o^N$' terms required in the VarQC equations provided by Andersson and Järvinen (1998).

*Competing interests.* The author has no competing interests

*Acknowledgements.* Niels Bormann, Peter Weston and Stephen English are thanked for very helpful discussions and reviews of the manuscript.





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
