# Peer review of "Correlated observation error models for assimilating all-sky infrared radiances"

_Atmospheric Measurement Techniques, 2018_

## Referee Comment (RC1) · Anonymous Referee #1 · 15 Dec 2018

Overall this was an excellent paper, a joy to read, and of significant practical value as well to operational NWP centers. The concepts presented will be applicable to other sets of channels and other instruments as well.

One relevant issue that I would like to see discussed is that of the sample eigenvalue spectrum of the observation error covariance matrix vs. the true eigenvalue spectrum. The expectation value of the largest eigenvalue is always overestimated by the sample eigenvalue, and the smallest is always underestimated. This provides additional justification for raising the value of the smallest sample eigenvalues. (This issue is discussed on a paper you reference, Campbell et al. 2017, which references Ledoit & Wolf, 2004)

Instead of the Desroziers method, which estimates R as the outer product of the departures and the obs minus analysis, this paper uses the outer production of the de-

partures with themselves, which yields HBHˆT + R. Because HBHˆT is small compared to R, especially in all-sky assimilation, this is justified; however, I would like to see a more quantitative estimate of the size of R relative to HBHˆT. One advantage is that for monitored observations, some opertional DA systems do not routinely produce obs minus analysis, which is an obstacle to the Desroziers calculation of R, but not to the calculation of HBHˆT + R.

Some minor comments:

P14, L20: Campbell et al. 2017 has an extensive discussion of trailing eigenvalues and condition number, so would be appropriate to include as a reference. P27, L15 Please provide a reference for the background fit to observations diagnostic presented here, and comments on how it compares to traditional forecast scores. P29, L11-12 Clarify how the blue curve is better than the orange curve P30 L15. They did not have to increase all evals; they chose to.

The remainder of my comments relate exclusively to the figures:

Fig 3. As noted in the text, eigenvectors are only unique up to sign, so the ones in this figure with the opposite sense should be multiplied by -1 and plotted, so as not to falsely draw the eye to a difference that is not real. Also the subfigures should be laid out differently to allow for larger size. A zero line would also be helpful.

Figs. 5-9, 16, 17, 21 Could use thicker lies to help differentiate the line color

Fig. 10. White should not be used to correspond both to a zero value and to unassimilated. Use e.g. gray over the Antarctic, Sahara, etc.

Fig 11. Colorbar does not need to extend to -2; -1 looks sufficient.

[Figure]

---

## Short Comment (SC1) · 14 Mar 2019

General comment:

This paper introduced an observation error model for correlated all-sky hyperspectral infrared sounders based on eigenvectors and corresponding eigenvalues, and presented methods to handle the problematic trailing eigenvalues that can cause unrealistic increments in the analysis when used as is. After reducing the sensitivity to the trailing eiginjacobians, the new error covariance matrix gives good results in all-sky infrared assimilation. This research is important as more potentially correlated observations are assimilated. The manuscript is well written and can be published after some minor modifications.

Specific comments: Eigen value decomposition mathematically finds the directions of largest variances within a dataset. While the leading eigenvalues and eigenvectors represents the majority of variance related to strong physical constrains and can be stable, the trailing ones may be sensitive to the training dataset used. The value of the trailing eigenvalues may be small, but it does not necessarily mean the error in the channel combinations represented by the trailing eigenvectors are small. Firstly, eigenvalue decomposition is a linear operation but radiative transfer under all-sky condition is highly non-linear. Secondly, the eigenvalue decomposition is optimized for the entire training dataset, but the Jacobians used in data assimilation is respect to the current model state. Since the leading eigenvalues are orders of magnitude larger than the trailing eigenvalues, any error 'leaks' from the leading 'eigenchannels' during data assimilation due to the aforementioned reasons can overwhelm the trailing eigenvalues. As such, the trailing eigenvalues should be trust less and should not be used directly. Maybe that's why these trailing eigenvalues should be inflated. The author may overstate the value of the trailing eigenvalues too much in the conclusion section (e.g., Page 41, Line 5-7) and suggest modifications to address the possible uncertainties when using the trailing eigenvalues.

Minor correction: Page 39, line 11: an extra 'because'

---

## Short Comment (SC2) · 12 Apr 2019

General Comments:

This paper presented an observation error model that combines the inter-channel correlations with the situation dependency as a function of symmetric cloud proxy variable required for the all-sky assimilation. This might be the first reported application of the correlated errors to the all-sky assimilation that provides the benefits to both the analysis and the NWP forecast accuracy. The need to inflate the trailing eigenvalues has been clearly explained through the concept of the eigendeparture and eigenjacobian, and the manuscript is well written.

Specific comments:

[Figure]

In the eigenspace spanned by the eigenvectors, the eigenvalues of the error covariance matrix are the equivalents of the error standard deviations which can be seen from the expression (6) on P9, so smaller the eigenvalues, larger the weights given to the eigendepartures. In this sense, the robustness of the error covariance matrices estimated based on different data samples and different version of systems should be assessed not only by the leading eigenvalues and eigenvectors but also by the trailing eigenvalues and eigenvectors. Both Fig.3 and Fig.4 indicate that the matrices examined have the relatively large differences in their trailing eigenvalues and eigenvectors. The data assimilation system might be very much sensitive and behave different because of these differences. Therefore, I suggest to run an additional experiment on top of the experiment "All-sky adjusted 1.0" or "All-sky adjusted 0.37" with any 45r1 all-sky error covariances to verify the robustness of the original 43r1 covariances that were used in all of the cycle experiments presented in this paper. My concern is the estimates might not be as robust as they look like in the sense that the extra tuning by trial and error might be still needed whenever to upgrade to a newer version of the matrix.

Technical corrections

P30L15, P33L8 and P41L25: all 'eigenvectors' should be replaced with 'eigenvalues'

---

## Referee Comment (RC2) · Anonymous Referee #2 · 19 Apr 2019

A very well-written manuscript with no obvious flaws which presents a clear picture of some experiments towards infrared all-sky radiance assimilation. The need to inflate the trailing eigenvalues and the justification for doing so is a significant finding and results in a consistency with much of the other work being done and what has been diagnosed from them for similar activities. I find the manuscript is ready for publication after corrections of any wording or clarity flaws which may be uncovered, but none were found by this reviewer.

I was very interested in one particular aspect of the paper. On page 30, beginning about line 20 when the trailing eigenvalues are adjusted so they are no smaller than 1 or 0.37. What is the resulting observation error in brightness temperature space for the clear-sky conditions as compared to the current "clear-sky" technique? Does

this result in high errors for these same clear-sky scenes when the all-sky technique is applied? It could be a very appropriate thing to do, and could even be indicating additional uncertainty should be added due to non-detection of partially cloud filled pixels.

Lastly, a small note the figures which use 2D line plots use very fine lines. This makes it particularly difficult to often discern between colors particularly the blue and black. Thicker lines though causing some overlap would make these much easier to discriminate.

Very last, very pithy comment. The label "all-sky diag" in figure 16 and 17 one could go ahead and spell out "diagonal" fully as there seems to be plenty of space for this in the figure label.
* * *

---

## Author Comment (AC1) · 3 May 2019

*Overall this was an excellent paper, a joy to read, and of significant practical value as well to operational NWP centers. The concepts presented will be applicable to other sets of channels and other instruments as well.*

The reviewer's positive and helpful comments are much appreciated.

**1.1** *One relevant issue that I would like to see discussed is that of the sample eigenvalue spectrum of the observation error covariance matrix vs. the true eigenvalue spectrum. The expectation value of the largest eigenvalue is always overestimated by the sample eigenvalue, and the smallest is always underestimated. This provides additional justification for raising the value of the smallest sample eigenvalues. (This issue*

[Figure]

*is discussed on a paper you reference, Campbell et al. 2017, which references Ledoit and Wolf, 2004)*

It is an interesting point that sample eigenvalues are more dispersed than those of the true covariance matrix, as explained by Ledoit Wolf (2004). However the severity of the problem decreases as $\frac{p}{n}$ decreases. Here, $p$ is the number of variables and $n$ the number of observations. In the current work $p$ is 7, for the 7 IASI channels, and $n$ is always substantially greater than 10,000. Hence the number of samples may be sufficient to minimise this problem from a purely statistical point of view. A point raised by other reviewers may be equally or more important here: is it really appropriate to use the same covariance matrix globally, across different seasons, across different model versions? It might be particularly the trailing eigenvectors that would vary in different conditions.

Mansuscript change: This point will be added to the discussion of eigenvector and eigenvalue stability at the end of section 3.2, with reference to Campbell et al. (2017) and Ledoit and Wolf (2004)

**1.2** *Instead of the Desroziers method, which estimates $R$ as the outer product of the departures and the obs minus analysis, this paper uses the outer production of the departures with themselves, which yields $HBH^T + R$. Because $HBH^T$ is small compared to R, especially in all-sky assimilation, this is justified; however, I would like to see a more quantitative estimate of the size of $R$ relative to $HBH^T$. One advantage is that for monitored observations, some operational DA systems do not routinely produce obs minus analysis, which is an obstacle to the Desroziers calculation of R, but not to the calculation of $HBH^T + R$.*

A justification for $HBH^T$ being significantly smaller than $R$ is already made in a few ways in the introduction to the manuscript, based on prior work. On P2L32, reference is made to Geer and Bauer (2011) whose tuning exercise suggested that all error in cloudy skies could be assigned to $R$, and to Harnisch et al. (2016) who showed ensemble estimates of $HBH^T$ around a third the size of $HBH^T + R$ estimated from the background departures. Further, as described in the introduction it has not been possible to provide similar estimates of $HBH^T$ from ECMWF's ensemble of data assimilations, because the recorded estimates appear to be affected by a bug that has not yet been solved. I would ask that further quantification of $HBH^T$ and $R$ be left to future work, as it will be a major exercise both technically and scientifically. Nevertheless this would be an important area on which to make progress

Manuscript change: The manuscript already justifies qualitatively that $HBH^T$ is significantly smaller than $R$ in all-sky conditions. It is agreed there is need to document this quantitatively, but it will be left for future work

*Some minor comments:*

**1.3** *P14, L20: Campbell et al. 2017 has an extensive discussion of trailing eigenvalues and condition number, so would be appropriate to include as a reference.*

Manuscript change: This citation will be added as requested at P14L20, and it will also now be covered in section 4.1 where the adjustment of eigenvalues in Weston et al. (2014) and Bormann et al. (2016) have been explained in more depth, but not as yet the Campbell et al. (2017) adjustments.

**1.4** *P27, L15 Please provide a reference for the background fit to observations diagnostic presented here, and comments on how it compares to traditional forecast scores.*

Unfortunately there is no good reference for these background fit diagnostics, but they are widely used at ECMWF and increasingly in studies by other NWP centres. However the point is well taken that they are used with little introduction in the current manuscript.

Manuscript change: Text will be added to describe the widespread use of such diagnostics, to justify them as an easy way of verifying the short-range forecast against observations, making use of diagnostics already generated from the DA process. These are now used in preference to short-range verification against analysis (whether the

experiment's own or from other analyses) because of the substantial error correlations between the forecast and analysis at short range, as demonstrated in e.g. Geer and Bauer (2010); Geer et al. (2010). Further, it will be pointed out that the change in the analysis fit should be used with care as a measure of analysis quality, as the observational reference is not independent of the analysis, but at least in this work the changes in analysis fits are consistent with the changes in background fits, where the reference observations are independent of the forecast.

**1.5** *P29, L11-12 Clarify how the blue curve is better than the orange curve*

Admittedly the effect of switching on VarQC is small, but there is significant impact in the background fits to ATMS channels 18, 19 and 20.

Manuscript change: Text will be added to clarify this, and the effect of VarQC will be more clearly signposted as minor both here and in the conclusion.

**1.6** *P30 L15.* They did not have to increase all evals; they chose to.

Manuscript change: "Have to" will be changed to "Chose to"

*The remainder of my comments relate exclusively to the figures:*

**1.7** *Fig 3. As noted in the text, eigenvectors are only unique up to sign, so the ones in this figure with the opposite sense should be multiplied by -1 and plotted, so as not to falsely draw the eye to a difference that is not real. Also the subfigures should be laid out differently to allow for larger size. A zero line would also be helpful.*

This figure is intended to occupy one column of the two-column A4 format used by AMT for final publication. It will appear slightly larger in that format when printed, and I believe of sufficient size to distinguish the different lines, but please let me know if that is not good enough.

Manuscript change: A zero line will be added and eigenvectors that differ only by sign will be replotted as suggested.

[Figure]

**1.8** *Figs. 5-9, 16, 17, 21 Could use thicker lies to help differentiate the line color.*

Manuscript change: I will experiment with thicker lines in these plots; this was a comment from one of the other reviewers too.

**1.9** *Fig. 10. White should not be used to correspond both to a zero value and to unassimilated. Use e.g. gray over the Antarctic, Sahara, etc.*

Manuscript change: I worked out how to get cross-hatching on these plots for the companion paper, so I will add that to Fig. 10 and Fig. 11 to distinguish areas where no data is used (such as over Antarctica).

**1.10** *Fig 11. Colorbar does not need to extend to -2; -1 looks sufficient.*

I will experiment if the scale range can be shrunk. However there are values up to around +1.6 in panel f, and it is necessary to have a symmetric contour range to retain blue colours for negative and red for positive. So it might be possible to reduce it as far as -1.5 to +1.5, but that requires further experimentation.

Manuscript change: The colorbar will be reduced in range if at all possible.

**References not in the original manuscript**

Geer, Alan J. and Peter Bauer, "Enhanced use of all-sky microwave observations sensitive to water vapour, cloud and precipitation.", ECMWF Technical Memorandum 620 (2010)

Geer, Alan J., Peter Bauer, and Philippe Lopez. "Direct 4D-Var assimilation of all-sky radiances. Part II: Assessment." Quarterly Journal of the Royal Meteorological Society 136, no. 652 (2010): 1886-1905.

Ledoit, Olivier, and Michael Wolf. "A well-conditioned estimator for large-dimensional covariance matrices." Journal of multivariate analysis 88, no. 2 (2004): 365-411.

---

## Author Comment (AC2) · 3 May 2019

*A very well-written manuscript with no obvious flaws which presents a clear picture of some experiments towards infrared all-sky radiance assimilation. The need to inflate the trailing eigenvalues and the justification for doing so is a significant finding and results in a consistency with much of the other work being done and what has been diagnosed from them for similar activities. I find the manuscript is ready for publication after corrections of any wording or clarity flaws which may be uncovered, but none were found by this reviewer.*

These positive and helpful comments are much appreciated

**2.1** *I was very interested in one particular aspect of the paper. On page 30, beginning*

[Figure]

*about line 20 when the trailing eigenvalues are adjusted so they are no smaller than 1 or 0.37. What is the resulting observation error in brightness temperature space for the clear-sky conditions as compared to the current "clear-sky" technique? Does this result in high errors for these same clear-sky scenes when the all-sky technique is applied? It could be a very appropriate thing to do, and could even be indicating additional uncertainty should be added due to non-detection of partially cloud filled pixels.*

This is an interesting point. The effect of eigenvalue adjustments on the clear-sky errors in the brightness temperature basis has been described in words but not visually, so a figure will be added, which would also support further discussion. As identified by the reviewer, the eigenvalue adjustment broadly increases observation error variances relative to those used in the clear-sky approach, but this is a broad change, not targeted at potentially cloudy areas, and at maximum the increase is around 20%. Bormann et al. (2016) tested a range of inflation factors for the error variances on the brightness temperature basis; they found that further error inflation beyond 1.75 (relative to their Desroziers estimate) did not improve the results. Hence to test the reviewer's hypothesis it would be interesting, although not feasible for the current manuscript, to try a more targeted error inflation by further inflating just the trailing eigenvalues of the clear-sky error covariance matrix in the clear-sky approach. However Campbell et al. (2017), in a clear-sky framework, found better convergence and fewer iterations to convergence using a "additive" adjustment to all eigenvalues, rather than the "Ky Fan" approach boosting only the trailing eigenvalues. This might hint that further trailing eigenvalue adjustment would not be that beneficial for clear-sky assimilation.

A likely more relevant aspect to improving the clear-sky approach is that many observations that are "clear" from the point of view of a "clear-sky" assimilation scheme are actually at locations where the model has cloud, often even deep convection. Though the clear-sky framework represents clear-sky radiative transfer in the observation operator, areas where the model is cloudy will have a high relative humidity, and the

observation equivalent may still be in substantial error compared to the observation that almost certainly comes from a clear area (the cloud detection schemes for hyper-spectral instruments seem to be very efficient). In the all-sky framework, these "obs clear - model cloudy" situations will get an additional boost to their observation error; however this effect is already present in the "baseline" all-sky experiment and is not further affected by eigenvalue adjustment.

Manuscript change: Add a new figure showing the effect of eigenvalue adjustment on the observation error in fully clear and fully cloudy situations. The possibility that error inflation in "clear" areas may provide part of the benefit of trailing eigenvalue adjustment will be acknowledged; however the clear-sky effect is likely a secondary explanation for the results.

**2.2** *Lastly, a small note the figures which use 2D line plots use very fine lines. This makes it particularly difficult to often discern between colors particularly the blue and black. Thicker lines though causing some overlap would make these much easier to discriminate.*

This was also mentioned by reviewer 1.

Manuscript change: I will experiment with thicker lines as advised.

**2.2** *Very last, very pithy comment. The label "all-sky diag" in figure 16 and 17 one could go ahead and spell out "diagonal" fully as there seems to be plenty of space for this in the figure label.*

Manuscript change: "Diagonal" will now be fully spelt out where it appears on the plots, and also in the text when the experiment is referred to by name.

---

## Author Comment (AC3) · 3 May 2019

*General comment: This paper introduced an observation error model for correlated all-sky hyperspectral infrared sounders based on eigenvectors and corresponding eigenvalues, and presented methods to handle the problematic trailing eigenvalues that can cause unrealistic increments in the analysis when used as is. After reducing the sensitivity to the trailing eigenjacobians, the new error covariance matrix gives good results in all-sky infrared assimilation. This research is important as more potentially correlated observations are assimilated. The manuscript is well written and can be published after some minor modifications.*

Thanks for these helpful and positive comments.

[Figure]

**3.1** *Specific comments: Eigen value decomposition mathematically finds the directions of largest variances within a dataset. While the leading eigenvalues and eigenvectors rep- resents the majority of variance related to strong physical constrains and can be stable, the trailing ones may be sensitive to the training dataset used. The value of the trailing eigenvalues may be small, but it does not necessarily mean the error in the channel combinations represented by the trailing eigenvectors are small. Firstly, eigenvalue de- composition is a linear operation but radiative transfer under all-sky condition is highly non-linear. Secondly, the eigenvalue decomposition is optimized for the entire training dataset, but the Jacobians used in data assimilation is respect to the current model state. Since the leading eigenvalues are orders of magnitude larger than the trailing eigenvalues, any error 'leaks' from the leading 'eigenchannels' during data assimilation due to the aforementioned reasons can overwhelm the trailing eigenvalues. As such, the trailing eigenvalues should be trust less and should not be used directly. Maybe that's why these trailing eigenvalues should be inflated.*

This point is well made and it is also supported in the comments by Wei Gu. Figure 3 does show that the detailed structure of the trailing eigenvectors, particularly for eigenchannel 6, does depend on the training dataset. It is hard to quantify the effect of these variations without an experiment like the one suggested by Wei Gu, which would re-run one of the existing experiments, but using error covariances from the 45r1 samples, to demonstrate the impact. I will definitely try this experiment but I would not like to commit to including the results in a revised manuscript due to the very long time it might take to complete.

Manuscript change: The stability of the trailing eigenvectors and values may be over-stated in my manuscript, and without the proposed experiment it is hard to quantify anyway. Hence the assertions of stability will be toned down in sections 3.2 and 4.1 in the revised manuscript. Also in the conclusion the possibility needs to be left open that trailing eigenvectors are not stable enough to justify putting high weights on them, and that this is also a possible explanation for the importance of inflating the trailing

eigenvalues, and that further work would be needed to investigate.

**3.2** *The author may overstate the value of the trailing eigenvalues too much in the conclusion section (e.g., Page 41, Line 5-7)*

Manuscript change: certainly the difference between the 0.37 and 1.0 eigenvalue adjustments is small, and this can be toned down on P41 L5-7 as suggested.

**3.3** *and suggest modifications to address the possible uncertainties when using the trailing eigenvalues.*

This is already addressed in the response to point 3.1 above.

**3.4** *Minor correction: Page 39, line 11: an extra 'because'*

Manuscript change: This typo will be corrected.

---

## Author Comment (AC4) · 3 May 2019

*General Comments: This paper presented an observation error model that combines the interchannel correlations with the situation dependency as a function of symmetric cloud proxy variable required for the all-sky assimilation. This might be the first reported application of the correlated errors to the all-sky assimilation that provides the benefits to both the analysis and the NWP forecast accuracy. The need to inflate the trailing eigenvalues has been clearly explained through the concept of the eigendeparture and eigenjacobian, and the manuscript is well written. Specific comments:*

These positive comments are very helpful and much appreciated.

**4.1** *In the eigenspace spanned by the eigenvectors, the eigenvalues of the error co-*

[Figure]

*variance matrix are the equivalents of the error standard deviations which can be seen from the expression (6) on P9, so smaller the eigenvalues, larger the weights given to the eigendepartures. In this sense, the robustness of the error covariance matrices estimated based on different data samples and different version of systems should be assessed not only by the leading eigenvalues and eigenvectors but also by the trailing eigenvalues and eigenvectors. Both Fig.3 and Fig.4 indicate that the matrices examined have the relatively large differences in their trailing eigenvalues and eigenvectors. The data assimilation system might be very much sensitive and behave different because of these differences. Therefore, I suggest to run an additional experiment on top of the experiment "All-sky adjusted 1.0" or "All-sky adjusted 0.37" with any 45r1 all-sky error covariances to verify the robustness of the original 43r1 covariances that were used in all of the cycle experiments presented in this paper. My concern is the estimates might not be as robust as they look like in the sense that the extra tuning by trial and error might be still needed whenever to upgrade to a newer version of the matrix.*

This is an important point and supported also by Fuqing Zhang. It is worrying that there seems to be such a need for trial and error retuning of error variances and eigenvalues, and possibly we might have to keep on doing this after significant model or observation changes. The proposed experiment is a good idea and clearly it is to hard to quantify what size and type of differences in the trailing eigenstructures would be important without experiment. However this might be the tip of the iceberg, and the issue of eigenstructure stability may deserve substantial further work beyond what can be included in the current manuscript. Further, the proposed experiment will take a long time to run so I would not want to commit to including it in a revised manuscript. I would start the experiment with the hope of reporting the results later. Hence the following proposal:

Manuscript change: Further discuss the stability of trailing eigenstructures in Figs. 3 and 4. In the conclusion call for further work investigating the stability, and acknowledge this remains another possible explanation for the benefit of adjusting the trailing eigenvalues.

*Technical corrections*

**4.2** *P30L15, P33L8 and P41L25: all 'eigenvectors' should be replaced with 'eigenvalues'*

Manuscript change: These issues will be fixed
* * *

---

## Author Response (AR1)

The main response to referees has been provided as part of the interactive discussion. This response contains the marked-up "diff" manuscript, with the notes below cross-referencing the responses from the interactive discussion to their final implementation in the "diff" manuscript. A few other small changes simply improve the clarity of the manuscript or correct small errors, in particular an error has been corrected (in two places on page 7) relating to the way that IASI thinning is done. Thanks again to the reviewers for their constructive and very helpful suggestions.

- 1.1 Eigenvector stability and sampling: Page 15 lines 6 to 12, additional discussion in conclusion

- 1.2 It was argued that the manuscript already contains sufficient justification

- 1.3 Expanded coverage of Campbell et al. (2017) on pages 15, 31, 33, 43

- 1.4 Expanded introduction to the ATMS fits plot on page 29

- 1.5 VarQC results justified better and toned down, page 30

- 1.6 Line 31-15

- 1.7 Figure 3 adjusted, text changed on page 14

- 1.8 All the colour line plots now have 50% thicker lines and the plots have been given more space on the page.

- 1.9 Cross-hatching now indicates missing data on Figs. 10 and 11.

- 1.10 Colourbar adjusted to -1.5 to +1.5 on Fig. 11

- 2.1 Figure 17 added, text changed on page 31

- 2.2 As 1.8

- 2.3 Fixed on page 31, table 3 and Fig. 18

- 3.1 Claims of eigenvector stability have been toned down on p. 16. However, the case is now made on page 15 that sampling errors may not be the main problem. A new summary paragraph in the conclusion, page 42, acknowledges the issue of eigenvector stability and calls for furher work, but argues that it may not be the main issue. New upublished results from all-sky microwave research are mentioned, where an experiment similar to reviewer 4's suggestion is now being carried out, but the early results suggest no major effect.

- 3.2 Page 43 lines 2 to 10 slightly tone down the arguments in favour of using the information from the trailing eigenvectors.

- 3.3 As 3.1

- 3.4 Page 41 line 5

- 4.1 Covariance (and eigenvalue/vector) stability is now more thoroughly covered on pages 15 and 16 and there is a new paragraph in the conclusion. See also responses to 1.1 and 3.1.

- 4.2 Corrections on pages 31, line 15, 34 line 18, 43 line 28.

[revised manuscript text omitted]